# Erk regulation of actin capping and bundling by Eps8 promotes cortex tension and leader bleb-based migration

Jeremy S Logue[1,2], Alexander X Cartagena-Rivera[2], Michelle A Baird[1], Michael W Davidson[3], Richard S Chadwick[2]*, Clare M Waterman[1]*

[1]National Heart Lung and Blood Institute, National Institutes of Health, Bethesda, United States; [2]National Institute on Deafness and other Communication Disorders, National Institutes of Health, Bethesda, United States; [3]National High Magnetic Field Laboratory and Department of Biological Science, Florida State University, Tallahassee, United States

**Abstract** Within the confines of tissues, cancer cells can use blebs to migrate. Eps8 is an actin bundling and capping protein whose capping activity is inhibited by Erk, a key MAP kinase that is activated by oncogenic signaling. We tested the hypothesis that Eps8 acts as an Erk effector to modulate actin cortex mechanics and thereby mediate bleb-based migration of cancer cells. Cells confined in a non-adhesive environment migrate in the direction of a very large 'leader bleb.' Eps8 bundling activity promotes cortex tension and intracellular pressure to drive leader bleb formation. Eps8 capping and bundling activities act antagonistically to organize actin within leader blebs, and Erk mediates this effect. An Erk biosensor reveals concentrated kinase activity within leader blebs. Bleb contents are trapped by the narrow neck that separates the leader bleb from the cell body. Thus, Erk activity promotes actin bundling by Eps8 to enhance cortex tension and drive the bleb-based migration of cancer cells under non-adhesive confinement.

*For correspondence:
chadwick@nidcd.nih.gov (RSC);
watermancm@nhlbi.nih.
gov (CMW)

Competing interests:
See page 28

## Introduction

Cell migration mediates critical physiological processes including development and the immune response, and is de-regulated during cancer metastasis. Cells move by orchestrating their extracellular interactions with intracellular cytoskeleton-dependent changes in shape and force generation; protruding forward, grasping sites in the environment, and pulling themselves along (*Lauffenburger and Horwitz, 1996*; *Papusheva and Heisenberg, 2010*). Although the pulling forces driving cell movement are well known to be mediated by myosin-based contraction (*Vicente-Manzanares et al., 2009*), recently it has become clear that migrating cells can utilize multiple mechanisms to drive protrusion and interaction with their environment (*Lammermann and Sixt, 2009*). Protrusion of the cell's plasma membrane boundary can be driven by either actin polymerization or by pressure-driven membrane blebbing (*Charras and Paluch, 2008*), while grasping the environment can be mediated by specific adhesion receptors that bind to the extracellular matrix (ECM) or other cells, or by non-specific friction (*Lammermann and Sixt, 2009*; *Bergert et al., 2015*). These multiple modes of migration are adopted by different cell types in different contexts. For example immune cells utilize adhesion-independent migration during tissue surveillance (*Lammermann et al., 2008*), while endothelial cells utilize polymerization-driven protrusion during angiogenesis (*Lamalice et al., 2007*).

Recent evidence indicates that physical confinement in a non-adhesive environment may drive a change of migration modes. In particular, contractility and extracellular pressure can drive a switch

**eLife digest** Cells within an animal have to be able to move both during development and later stages of life. For example, white blood cells have to move around the body and into tissues to fight off infections. Normally, cell movement is heavily controlled and will only happen when it is necessary to keep an animal healthy. However, cancer cells can bypass these controls and 'metastasize', or spread to new sites in the body.

Cells can move in several different ways: on the one hand, cells can use 'mesenchymal' movement, in which the skeleton-like scaffolding of molecules within a cell rearranges to push the cell forward. On the other hand, cells can employ 'amoeboid' movement, in which they squeeze their way forward by building up pressure in the cell. Although these different types of movement are only used by some healthy cells and not others, cancer cells can switch between the two. How they do this is still unclear, but now Logue et al. have studied this question using several microscopy techniques.

Logue et al. watched skin cancer (or melanoma) cells migrating between a glass plate and a slab of agar, which mimics the confined spaces that cancer cells have to move through within the body. The images showed that the cancer cells formed so-called 'leader blebs', finger-like projections that put cells on the right track. The experiments revealed that a protein called Eps8 was responsible for making the skin cancer cells move in this amoeboid fashion. The 'blebbing' caused by Eps8 is turned on by another protein called Erk that is often overactive in melanoma cells. Furthermore, Erk can accumulate near and within the cell blebs and this leads to the increased movement of the skin cancer cells.

Studying cell movement in melanoma is particularly important because it is the metastatic tumors that kill patients. Therefore, increasing our basic understanding of how cells migrate could eventually lead to better treatment options that stop cancer cells from spreading.

from polymerization/adhesion-based to bleb/friction-based motilities known as the mesenchymal-to-amoeboid transition (MAT) (*Bergert et al., 2015*; *Liu et al., 2015*; *Ruprecht et al., 2015*). Cancer cells are known to be highly contractile, making them prone to blebbing (*Bergert et al., 2012*), while tumors are known to be sites of high turgor pressure (*Jain, 1987*), both properties being conducive to MAT. Indeed, intravital imaging revealed that melanoma and breast cancer cells migrate by blebbing in live mice (*Tozluoglu et al., 2013*). Thus, metastatic cancer cells are hypothesized to be susceptible to MAT in vivo, with the plasticity of their migration modes and lack of specificity in adhesion contributing to their high invasivity and difficulty targeting (*Lammermann and Sixt, 2009*).

Polymerization-driven mesenchymal migration and bleb-based amoeboid migration are mechanistically distinct. In mesenchymal migration, actin polymerization initiated by localized activation of a filament-nucleating factor drives the formation of actin networks or bundles that mediate lamellipodial or filopodial membrane protrusion (*Skau and Waterman, 2015*). In contrast, in blebbing cells, membrane protrusion is mediated by myosin II contractility-induced hydrostatic pressure that drives a bubbling-out of the plasma membrane at a site of local weakness in the cortical actin cytoskeleton (*Charras et al., 2005*; *Charras and Paluch, 2008*). While the molecular mechanisms mediating polymerization-based protrusion and contractility-induced pressure are reasonably well understood, the molecules responsible for local changes in the actin cortex that allow bleb formation are not known. It is hypothesized that blebs could form either by local down-regulation of membrane-cytoskeleton linkers such as those of the ezrin/radixin/moesin family (*Estecha et al., 2009*; *Lorentzen et al., 2011*), or by local weakening of sites in the cortical actin network itself (*Charras et al., 2006*). Indeed, experiments using actin depolymerizing drugs have shown cortex integrity to be an important factor in regulating blebbing (*Charras et al., 2006*). Local inhomogeneity in cortical integrity could be mediated by regulation of actin filament number or organization. In turn, filament number could be controlled by modulating the activity of filament nucleators, cappers or depolymerizers, while actin organization could be regulated by filament crosslinkers, bundlers or motor proteins. However, the proteins critical to regulating actin cortical organization and mechanics during blebbing are not known.

In this study, we focused on identifying the mechanism for regulating the actin cortex of cancer cells during MAT and bleb-based migration. Many highly invasive cancers are known to be caused by

mutations that activate the oncogenic EGF/Ras/Raf/MEK/Erk pathway (*Downward, 2003*; *Roberts and Der, 2007*). It is well-known that Erk-mediated activation of myosin II promotes cell contractility (*Klemke et al., 1997*). Accordingly, we concentrated on actin regulatory proteins downstream of this pathway that may synergize with contractility to affect blebbing. Epidermal growth factor receptor pathway substrate 8 (Eps8) is an actin bundling and capping protein (*Disanza et al., 2004*, *2006*; *Hertzog et al., 2010*) that plays a critical role in development of the nervous, auditory and reproductive systems (*Lie et al., 2009*; *Menna et al., 2009*, *2013*; *Manor et al., 2011*) and its upregulation in cancers correlates with invasivity and poor prognosis (*Griffith et al., 2006*; *Wang et al., 2009*; *Kang et al., 2012*). Eps8 was originally identified as an actin binding protein downstream of the EGF receptor and to regulate actin through a complex with SOS1 and Abi1 (*Fazioli et al., 1993*; *Scita et al., 1999*). Additionally, the actin filament capping activity of Eps8 is inhibited by Erk through phosphorylation (*Menna et al., 2009*). Thus, the dual actin regulatory functions and targeting by Erk make Eps8 a good candidate for regulating the cortex downstream of oncogenic signaling.

By combining imaging of fluorescently tagged proteins and atomic force microscopy (AFM) with targeted mutations in Eps8, the work described here tests the hypothesis that Eps8 acts as a key effector of Erk to modulate actin cortex mechanics and thereby mediate bleb-based migration of cancer cells. We find that Eps8 bundling activity promotes cortex tension and pressure to drive MAT in non-adherent, confined cells, which migrate with a characteristic 'leader-bleb' morphology. Within leader blebs, Eps8 capping and bundling activities act antagonistically, and phosphorylation by Erk mediates this effect. Using a FRET biosensor for Erk, we document a massive concentration of kinase activity within leader blebs and find that leader bleb contents are trapped by the bleb neck that separates the bleb from the cell body. Our results identify a mechanism by which Eps8 may promote the transition to rapid, unregulated migration of cancer cells in confinement that may be critical to their highly invasive behavior in vivo.

## Results

### Eps8 is recruited early to bleb membranes and forms a gradient across the length of a 'leader bleb'

To determine the role of Eps8 in cancer cell blebbing and migration, we utilized human A375 melanoma cells which carry a mutation in B-Raf (V600E) that activates the Raf/MEK/Erk pathway (*Davies et al., 2002*). We imaged Eps8 and the actin cytoskeleton by spinning disk confocal microscopy in cells that were fixed and stained with fluorescent phalloidin and either immunolabeled with antibodies to Eps8 or expressing Emerald-tagged mouse Eps8 (Emerald-mEps8). When cells were plated on fibronectin-coated coverslips to promote adhesion and spreading, actin formed a dense meshwork in the lamellipodia near the cell edge, and circumferential arcs and stress fibers in the lamella and cell body (*Figure 1A*). Both endogenous and Emerald-mEps8 localized primarily to lamellipodia and arcs, but were absent from stress fibers (*Figure 1A*, *Figure 1—figure supplement 1A*). To determine Eps8 distribution in a non-adhesive environment, A375 cells co-expressing Emerald-mEps8 and FusionRed-tagged F-tractin (an actin filament binding peptide, [*Schell et al., 2001*]) were plated on uncoated glass. Here, cells were rounded, and confocal sections midway through the cell Z-axis revealed a dense band of cortical actin at the periphery of the cell body. From this cortical band extended many blebs that possessed thin, continuous rims of cortical actin at their membranes (*Figure 1D*). Eps8 was absent from the dense cortical band of the cell body, but localized with actin in punctae along the peripheral rim of actin in blebs (*Figure 1B,D*). Co-expression of FusionRed-F-tractin and EGFP-tagged myosin II regulatory light chain (GFP-MII-RLC, a marker of myosin II isoforms) in non-adherent cells showed that myosin II was concentrated on the dense cortical band and was at very low levels or absent from the bleb periphery (*Figure 1F*). Thus, Eps8 shows differential localization to actin structures in adherent spread and non-adherent blebbing melanoma cells.

We next sought to determine the dynamics of Eps8 in blebbing cells. Previous studies have shown that shortly after bleb protrusion, the membrane-cytoskeleton linker protein ezrin is recruited to the bleb membrane, followed by the assembly of actin, then myosin II, which induces bleb retraction (*Charras et al., 2006*). To determine the timing of Eps8 arrival at the bleb membrane relative to these proteins, we subjected A375 cells expressing EGFP tagged mEps8, ezrin, F-tractin, or MII-RLC to time-lapse confocal imaging at 5 s intervals. To mark the position of the cell membrane in negative-image, cells were mounted in media containing red fluorescent dextran (*Figure 1C,E,G*,

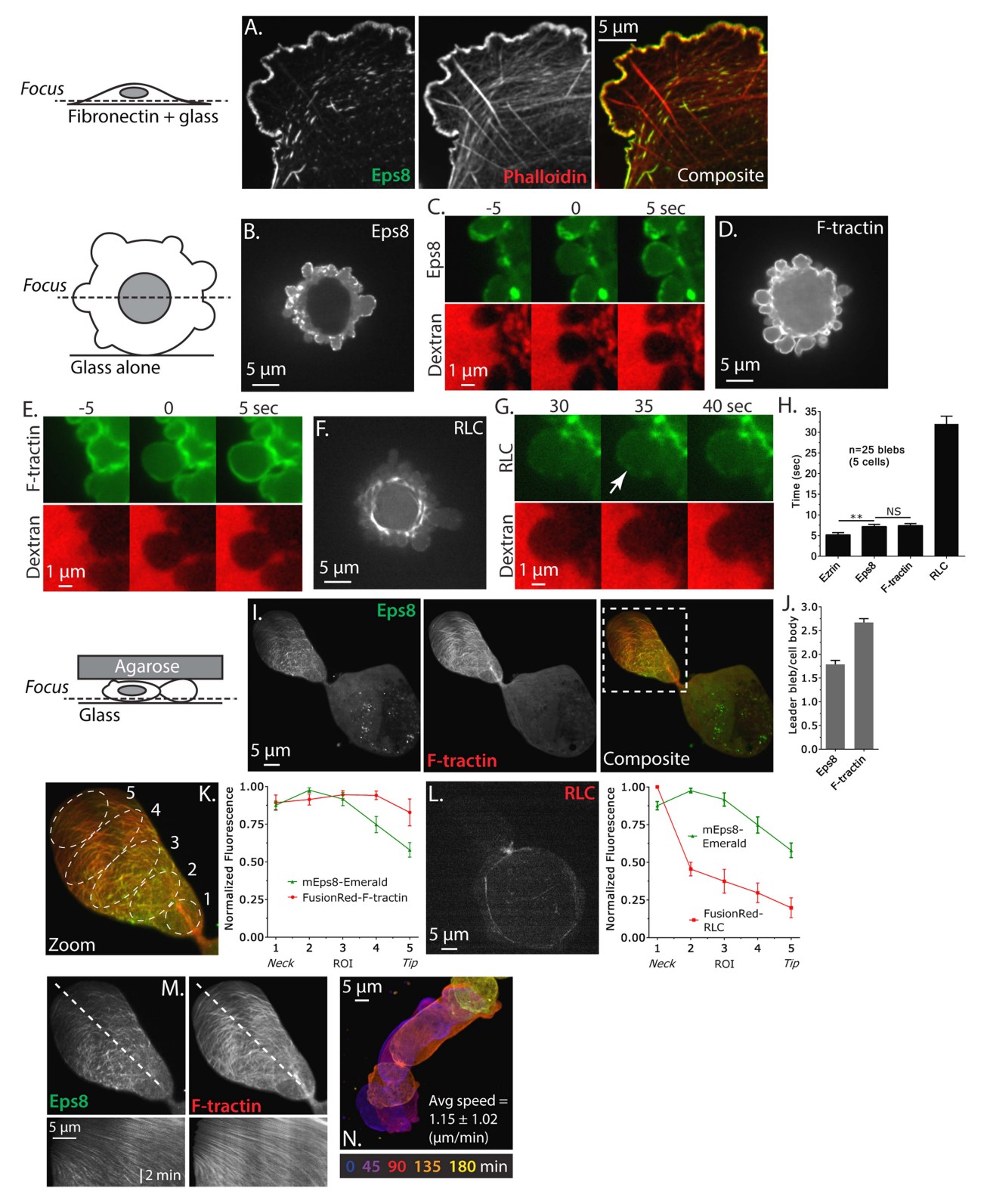

**Figure 1**. Eps8 is recruited early to bleb membranes and forms a gradient across the length of a 'leader bleb.' (**A**) Confocal image of the ventral Z-plane of human A375 melanoma cells expressing Emerald-tagged mouse Eps8 (green) plated on fibronectin-coated glass and stained with phalloidin (red). (**B–G**) Confocal images through the central Z-plane of A375 cells plated on uncoated glass. (**B**) Cell expressing Emerald-mEps8. (**C**) Time-lapse series of Emerald-mEps8 (green) and rhodamine-dextran (red) used as a negative stain in the culture media to detect the position of the cell boundary. Times in
*Figure 1. continued on next page*

*Figure 1. Continued*

(**C**, **E**, **G**) indicate seconds after the formation of a new bleb. (**D**) Cell expressing Emerald-tagged F-tractin to label actin filaments. (**E**) Time-lapse series of Emerald-F-tractin and rhodamine-dextran. (**F**) Cell expressing EGFP-tagged myosin II regulatory light chain (EGFP-RLC). (**G**) Time-lapse series of EGFP-RLC and rhodamine-dextran. (**H**) Quantification of the average time of appearance of EGFP or Emerald-tagged cortical proteins (EGFP-Ezrin, Emerald-mEps8, Emerald-F-tractin, and EGFP-RLC) relative to the time of maximal membrane protrusion determined from time-lapse series similar to those shown in (**C**, **E**, **G**). (**I–K–N**) Confocal images through the ventral Z-plane of A375 cells plated on uncoated glass and confined under an agar slab. (**I**) Cell co-expressing Emerald-mEps8 (green) and FusionRed-F-tractin (red). Boxed area is shown zoomed in (**K**). (**J**) Average ratio of fluorescence in the leader bleb to that in the cell body for Emerald-mEps8 and FusionRed-F-tractin. (**K**) (Left) Example of 5 regions of interest (ROIs), each 20% of the length of the leader bleb, used for (Right) regional analysis of the average fluorescence (normalized to maximum) of Emerald-mEps8 (green) and FusionRed-F-tractin (red) along leader blebs. (**L**) (Left) Image and (right) regional analysis of FusionRed-RLC (red) and Emerald-mEps8 (green, image not shown) fluorescence in leader blebs. (**M**) (Top) Image showing the position (dotted line) along which kymographs (bottom) of Emerald-mEps8 and FusionRed-F-tractin were made from time-lapse videos of their dynamics in leader blebs. Scale bar: 2 min (**N**) Color encoded time-overlay of images of Emerald-F-tractin in a migrating cell. Error is SEM, *p ≤ 0.05, **p ≤ 0.01, ***p ≤ 0.001, ****p ≤ 0.0001, NS: p > 0.05. See also *Videos 1–3*.

The following figure supplements are available for figure 1:

**Figure supplement 1**. Eps8 and ezrin localization.

**Figure supplement 2**. Detailed view of Eps8 and actin localization.

*Figure 1—figure supplement 1B*). Analysis of time-lapse image series showed that similar to previous studies, ezrin appeared rapidly on the newly protruded bleb membrane at ∼5 s after bleb formation (*Figure 1H*, *Figure 1—figure supplement 1B*). Recruitment of Eps8 and actin to the membrane occurred on a similar timescale as ezrin (*Figure 1C,E,H*). In contrast, MII-RLC appeared ∼30 s after protrusion, and coincided with the onset of bleb retraction (*Figure 1G,H*). These results indicate that in non-adherent cells, Eps8 recruitment to bleb membranes occurs concurrently with actin assembly.

To induce migration of non-adherent melanoma cells, we confined A375 cells expressing Emerald-mEps8 and FusionRed-F-tractin between an agarose pad and uncoated glass (*Bergert et al., 2012*). Strikingly, cells in these conditions had reduced blebbing on the cell body, but formed a single very large sausage-shaped bleb that generally excluded the nucleus, and which was defined by a thin neck at the junction between the large bleb and the spherical cell body (*Figure 1I* and *Video 1*). This morphology was very similar to the 'A2' or 'stable bleb' phenotypes recently described for non-adherent cells migrating under confinement (*Bergert et al., 2015*; *Liu et al., 2015*; *Ruprecht et al., 2015*). Long-term imaging showed that ∼40% of non-adherent, confined cells exhibited apparently rapid migration in the direction of the very large bleb, which we will thus call a 'leader bleb' (*Video 2*). Indeed, A375 cells adhered to fibronectin-coated coverslips (either 5 or 50 μg/ml) did not migrate, while leader bleb-based migration of confined non-adherent cells was very fast at 1.15 ± 1.02 μm/min (mean ± SE, *Figure 1N*).

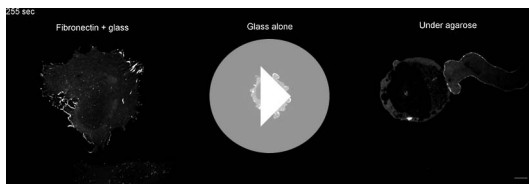

**Video 1.** Human melanoma A375 cells form a single prominent bleb when confined under an agar pad. Comparison of central Z-plane confocal time-lapse videos of Emerald-mEps8 dynamics in A375 cells plated on human plasma fibronectin coated glass (left), un-coated glass (middle) and confined between uncoated glass and an agar pad (right). Scale bar: 5 μm, elapsed time in seconds shown.

Confocal imaging revealed that both Eps8 and F-actin were concentrated within the leader bleb relative to the cell body (*Figure 1J*, *Figure 1—figure supplement 1C*, *Figure 1—figure supplement 2*). To analyze the spatial distribution of actin and Eps8 within the leader bleb, we determined their average intensities within five regions of interest (ROIs), each representing 20% of the length of the leader bleb from neck (region 1) to distal tip (region 5) (*Figure 1K*). This showed that actin was distributed as circumferential bundles around the short axis of most of the length of the leader bleb, but was reduced at the bleb tip (*Figure 1I,J*, *Figure 1—figure supplement 1C*, *Figure 1—figure supplement 2*). Eps8 localized in a punctate manner along actin bundles, forming

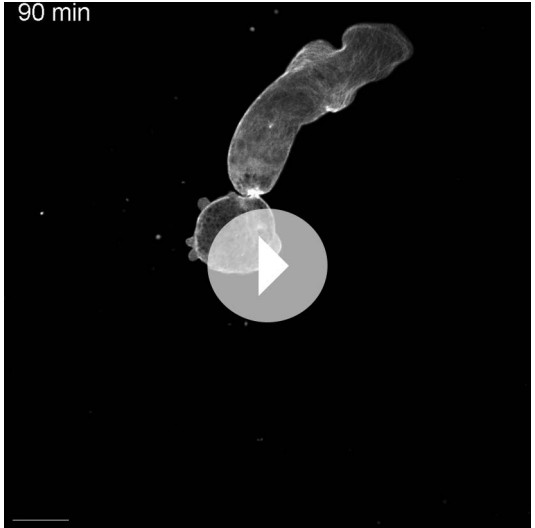

**Video 2.** Human melanoma A375 cells migrate in the direction of a 'leader bleb.' Central Z-plane confocal time-lapse video of Emerald-F-tractin showing actin dynamics during 'leader bleb' based migration of an A375 cell confined between uncoated glass and an agar pad. Scale bar: 5 µm, elapsed time in minutes shown.

**Video 3.** Eps8 and actin flow towards the bleb neck in leader blebs. Ventral Z-plane confocal time-lapse video of Emerald-mEps8 (green) and FusionRed-F-tractin (red) dynamics showing their coordinated flow towards the bleb neck in an A375 cell confined between uncoated glass and an agar pad. Scale bar: 5 µm, elapsed time in seconds shown.

a gradient, with the highest concentration near the neck connecting the bleb to the cell body and the lowest level at the distal tip of the leader bleb (*Figure 1K*, *Figure 1—figure supplement 1D*, *Figure 1—figure supplement 2*). Time-lapse imaging and kymograph analyses showed that Eps8 and F-actin underwent coordinated retrograde movement from the distal leader bleb tip towards the bleb neck (*Figure 1M* and *Video 3*). Similar imaging of EGFP-MII-RLC showed that myosin II was distributed around the cortical band of the cell body, highly concentrated in the narrow neck at the bleb neck where the bleb connected to the cell body, and nearly absent from the distal half of the leader bleb, similar to previous reports (*Figure 1L*) (*Bergert et al., 2015*; *Liu et al., 2015*; *Ruprecht et al., 2015*). Together, these results show that in non-adherent cells, Eps8 localizes rapidly to bleb membranes as actin assembles. When non-adherent cells are confined, actin and Eps8 concentrate in leader blebs where they exhibit a directional assembly gradient and rearward flow towards the contractile bleb neck, and this cortical flow is coordinated with rapid cell movement directed by the leader bleb.

## Eps8 promotes leader bleb-based migration in cancer cells by maintaining actin bundles towards the distal bleb tip

We next sought to determine the requirement for Eps8 in the promotion of leader bleb-based migration and organization of the cortical cytoskeleton in confined, non-adherent cells. To test this, we used a small interfering RNA (siRNA) targeted to human Eps8 that resulted in 76 ± 2.1% (mean ± SE) depletion of the protein in A375 cells after 24 hr (Eps8-KD, *Figure 2A*). Expression of FusionRed-F-tractin in non-adherent Eps8-KD cells confined under an agar pad followed by confocal imaging revealed that Eps8 depletion generally inhibited the formation of large leader blebs (*Figure 2B*). Instead, confined non-adherent cells resembled unconfined non-adherent cells, with a distribution of various-sized protruding and retracting blebs around their perimeter (*Figure 2B*), and the fraction of cells that underwent leader bleb-based migration was reduced by nearly half (*Figure 2D*). To quantify the effects of Eps8-KD on bleb size, we used a simple definition of a leader bleb as the single largest bleb made by the cell, expressed as a percent of cell body area. This showed that Eps8-KD decreased leader bleb area by nearly half compared to non-targeting siRNA (*Figure 2C* and *Supplementary file 1A*). Expression of Emerald-mEps8 fully rescued leader bleb size and migration in Eps8-KD cells (*Figure 2C–D*,

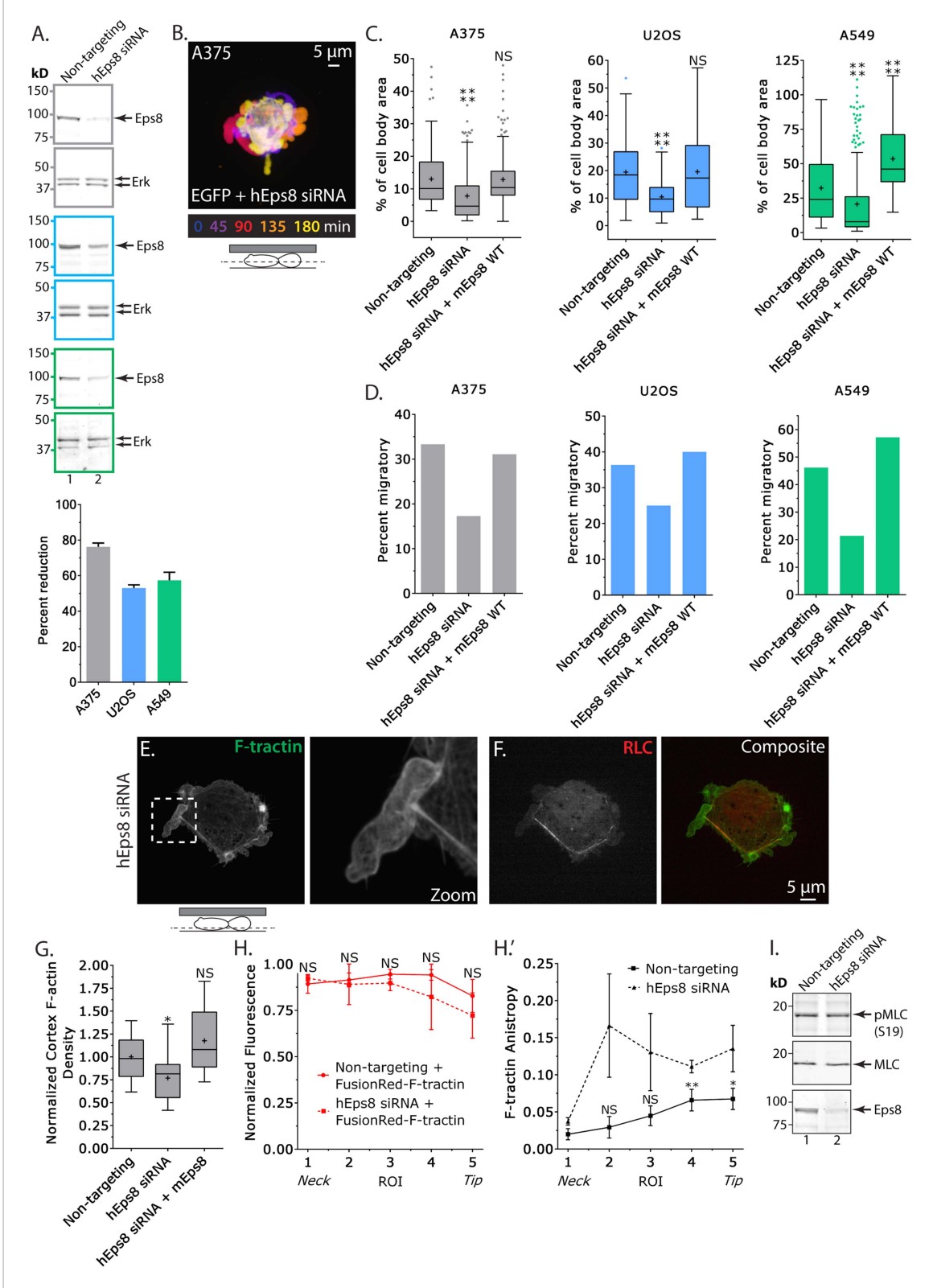

**Figure 2**. Eps8 promotes leader bleb-based migration by maintaining actin bundles towards the distal bleb tip. (**A**) (Left) Representative blot and (right) quantitation of Western blot analyses of Eps8 and Erk in lysates of A375 (gray), U2OS (blue) and A549 (green) cells that were treated with non-targeting siRNAs (non-targeting) or siRNAs targeting human Eps8 to deplete Eps8 (hEps8 siRNA). (**B–H**) Images and analyses of cells plated on glass and confined under an agar slab. (**B**) Color encoded time-overlay of confocal images through the central Z-plane of an A375 cell depleted of Eps8 and expressing

*Figure 2. continued on next page*

*Figure 2. Continued*

soluble EGFP. (**C, G**) Tukey box plots showing (**C**) quantification of leader bleb area expressed as a % of cell body area for cells treated with non-targeting or Eps8 siRNAs, with or without the additional expression of mouse Eps8 (mEps8 WT). '+' and line denote the mean and median, respectively. (**D**) Quantitation of the percent of cells that migrate from time-lapse phase contrast videos, treatments as in (**C**). (**E, F**) Confocal images through the ventral Z-plane of an A375 cell depleted of Eps8 and expressing Emerald-F-tractin (**E**) and FusionRed-myosin II regulatory light chain ((**F**), RLC), boxed area in **E** shown at right, overlay of (**E**) and (**F**) shown at left of (**F**). (**G**) Analysis of cortical actin density in A375 cells (see Materials and methods) in the cell body from images of phalloidin, treatments as in (**C**), normalized to the mean value of non-targeting control. (**H**) Regional analysis of the average fluorescence intensity (**H**, normalized to maximum) and bundle anisotropy (**H'**) of FusionRed-F-tractin along leader blebs in A375 cells treated with either non-targeting or human Eps8 siRNAs. Each point represents the average value in a region of interest (ROI) that is 20% of the length of the leader bleb. (**I**) Western blot analyses of Eps8, myosin II regulatory light chain (MLC), and myosin II regulatory light chain phosphorylated on serine 19 (pMLC (S19)) in lysates of A375 cells that were treated with non-targeting siRNAs (non-targeting) or siRNAs targeting human Eps8 to deplete Eps8 (hEps8 siRNA). Error in (**A, H**) is SEM, *$p \leq 0.05$, **$p \leq 0.01$, ***$p \leq 0.001$, ****$p \leq 0.0001$, NS: $p > 0.05$.

The following figure supplements are available for figure 2:

**Figure supplement 1**. Myosin II localization is unperturbed in human melanoma A375 cells depleted of and rescued with Eps8.

**Figure supplement 2**. Eps8 is required for leader bleb formation in A549 and U2OS cells.

*Figure 2—figure supplement 1C*, *Supplementary file 1A*). Thus, Eps8 promotes formation of a large leader bleb to drive migration of confined, non-adherent A375 cells.

We then sought to determine whether the requirement for Eps8 in leader bleb-based migration was a more general property of cancer cells or specific to A375 cells. We tested this in U2OS human osteosarcoma that are deleted for the cell cycle regulatory gene CDKN2A (Catalogue of somatic mutations in cancer), and human lung cancer A549 cells that carry the K-Ras G12S oncogenic mutation (Catalogue of somatic mutations in cancer). When plated on uncoated glass, both U20S and A549 cells rounded up and exhibited blebbing around their peripheries (*Figure 2—figure supplement 2*). When transfected with EGFP as a soluble marker and confined between uncoated glass and agar and subjected to time-lapse confocal microscopy, 36% of U20S cells and 46% of A549 cells took on a leader bleb morphology (*Figure 2—figure supplement 2*) and underwent rapid migration (*Figure 2D*). To test the requirement for Eps8 in this transition, we used siRNA to reduce Eps8 level by 53% and 57% in U20S and A549 cells, respectively (*Figure 2A*). Analysis of confocal videos showed that Eps8 depletion generally inhibited the formation of large leader blebs (*Figure 2—figure supplement 2*), and quantitation confirmed that the fraction of both cell types that underwent leader bleb-based migration was reduced by more than half, and leader bleb area was reduced by 34% and 43% in U20S and A549 cells, respectively (*Figure 2C,D*). Expression of Emerald-mEps8 rescued leader bleb size and migration in U20S and A549 cells that had been treated with Eps8 siRNA (*Figure 2C–D*, *Figure 2—figure supplement 2*, *Supplementary file 1A*). Thus, Eps8 is required for leader bleb formation to drive migration of confined, non-adherent cells in several cancer cell types, independent of the defect driving transformation.

To analyze the effects of Eps8-KD on cytoskeletal organization in A375 cells (*Figure 2E–H*), we utilized metrics that quantified both the density and bundle organization of the cytoskeleton in confined, non-adherent cells. For filament density, we measured average F-tractin intensity in both the dense cortical actin band of the cell body and in the leader bleb (*Figure 2G,H*). To quantify bundle organization, we determined the local alignment ('anisotropy') of F-tractin bundles within the leader bleb via the ImageJ plugin, FibrilTool, which uses the concept of nematic tensor to provide a local quantitative description of the anisotropy of fiber arrays and their average orientation in cells (*Boudaoud et al., 2014*) (*Figure 2H*). To quantify the spatial variation in these parameters, we determined their values in five ROIs along the leader bleb length (*Figure 2H*). This showed that compared to control, Eps8-KD induced a slight but significant reduction in the density of the cortical actin band in the cell body (*Figure 2E,G*), but had no significant effect on the density or distribution of actin within leader blebs, where actin exhibited a gradient with the highest concentration at the neck (*Figure 2E,H*). Regional analysis of F-tractin anisotropy showed that in control cells, actin bundle alignment also formed a gradient in the leader bleb, but in the opposite direction as the density gradient, such that bundles consistently were most highly aligned towards the distal regions

(*Figure 2H″*), and decreased towards the neck. Knockdown of Eps8 did not destroy the anisotropy gradient, but made actin bundle organization highly variable along the largest bleb (*Figure 2H″*). Examination of the localization of EGFP-MII-RLC in Eps8-KD cells showed that similar to control, myosin II concentrated in the dense cortical band of actin around the cell body and at the neck of the largest bleb (*Figure 2F*, *Figure 2—figure supplement 1B*). In agreement, western blotting for myosin II regulatory light chain phosphorylated on serine 19 (pS19 MLC) as a marker of activated myosin II showed no significant difference in myosin II activity between control and Eps8-KD (*Figure 2I*). Together, these results show that in confined, non-adherent cells, Eps8 promotes enlargement of leader blebs independent of myosin II activity, where it acts to promote actin bundling towards the distal bleb tip, and is required for leader bleb-based migration.

## Cortex tension and intracellular pressure are maintained by Eps8

Since leader bleb formation is mediated by intracellular pressure induced by contractility in the cytoskeleton (*Bergert et al., 2015*; *Liu et al., 2015*; *Ruprecht et al., 2015*), we next sought to determine the role of Eps8 in regulation of cortical cytoskeleton mechanical properties. We utilized an atomic force microscope (AFM)-based assay in which rounded, non-adherent cells are subjected to a minimal deformation with a tipless cantilever (*Figure 3A,B*) (*Fischer-Friedrich et al., 2014*; *Ramanathan et al., 2015*), and cortex tension and intracellular pressure can be extracted from the resulting force–displacement curves using measurements of the cell radius and actin cortex thickness

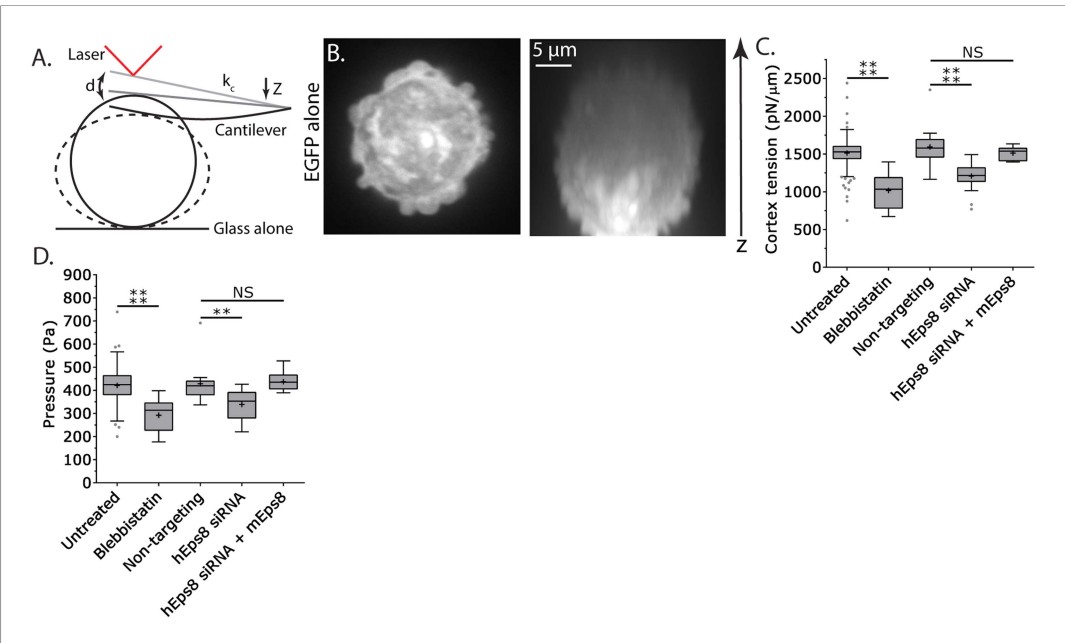

**Figure 3**. Cortex tension and intracellular pressure are maintained by Eps8. (**A**) Schematic representation of the Atomic Force Microscope (AFM) based assay for determining cortex tension and intracellular pressure in A375 cells plated on uncoated glass. 'k$_c$' cantilever spring constant, 'd' cantilever deflection, 'z' piezo Z displacement (**B**) (Left) confocal image of the central Z plane or (right) x-y projection of a 3D reconstruction of a Z-stack of a cell expressing EGFP as a volume marker. (**C**, **D**) Tukey box plots of cortex tension (**C**) and intracellular pressure (**D**) determined from AFM analysis. '+' and line denote the mean and median, respectively. Cells were treated with 50 μM blebbistatin, non-targeting siRNAs, siRNAs targeting human Eps8 (hEps8) with or without the additional expression of Emerald-tagged mouse Eps8 (mEps8). *p ≤ 0.05, **p ≤ 0.01, ***p ≤ 0.001, ****p ≤ 0.0001, NS: p > 0.05.

The following figure supplement is available for figure 3:

**Figure supplement 1**. Actin cortex thickness and cell radii are not significantly affected by depletion of and rescue with wild type Eps8 or its mutants.

(*Clark et al., 2013*) and a theory derived from a simple force balance of the applied cantilever normal force with the force due to intracellular pressure and the force from cortex tension (see 'Materials and methods'). We first validated the approach using cells treated with the myosin II ATPase inhibitor blebbistatin. This showed that inhibition of myosin II did not significantly affect cell radius or cortical actin thickness, but cortex tension and intracellular pressure were both significantly reduced (*Figure 3C–D*, *Figure 3—figure supplement 1*, *Supplementary file 1*), consistent with previous reports (*Tinevez et al., 2009*). Similar measurements showed that compared to non-targeting control, Eps8-KD resulted in significant decreases in cortex tension and intracellular pressure, although these effects were not as strong as those induced by blebbistatin (*Figure 3C–D*, *Figure 3—figure supplement 1*, *Supplementary file 1*). The effects of Eps8-KD on mechanical properties could be rescued by re-expression of Emerald-mEps8 (*Figure 3C–D*, *Figure 3—figure supplement 1*, *Supplementary file 1*). Thus, Eps8 promotes cortex tension and increases intracellular pressure. Together with our above results, this suggests that Eps8 regulates cortical cytoskeletal organization to enhance cortical tension and increase intracellular pressure to mediate leader bleb-based migration.

## Actin bundling by Eps8 promotes cortex tension and intracellular pressure to drive leader bleb formation

We next sought to test whether the actin bundling activity of Eps8 is required for leader bleb formation and migration, cytoskeletal organization, and cellular mechanical properties. To accomplish this, we made alanine substitutions within the C-terminus of Emerald-tagged mouse Eps8 (L757A/K759A, referred to as Emerald-mEps8Δbund, *Figure 4A*) which have been previously shown to specifically block Eps8 bundling activity (*Hertzog et al., 2010*). We co-expressed this together with FusionRed-F-tractin or FusionRed-MII-RLC in either wild type or Eps8-KD cells in a non-adherent, confined environment. This showed that Emerald-mEps8Δbund co-localized with actin at the thin cortical rim on bleb membranes, similar to wild type Eps8 (*Figure 4B,F*, *Figure 4—figure supplement 1*). However, in confined Eps8-KD cells, expression of Emerald-mEps8Δbund failed to rescue the defect in leader bleb size induced by loss of Eps8, and these cells remained round and exhibited small blebs around their periphery with no dominant leader bleb (*Figure 4B–C*, *Figure 4—figure supplement 1*, *Supplementary file 1A* and *Video 4*). Accordingly, Eps8-KD cells expressing Emerald-mEps8Δbund exhibited a more than 50% decrease in the fraction of cells migrating under confinement compared to Eps8-KD cells reconstituted with Emerald-mEps8 (*Figure 4E*). Even stronger effects on leader bleb size were observed in wild-type A375 cells over-expressing Emerald-mEps8Δbund (*Figure 4D* and *Supplementary file 1B*), indicating it acts as a dominant negative, likely by dimerizing with endogenous Eps8 (*Kishan et al., 1997*). We thus used a dominant negative approach where appropriate to negate the possibility of off-target effects induced by siRNAs. Because cells carrying defects in Eps8 bundling activity lacked a large bleb, we were unable to determine the effects of this mutation on actin density or anisotropy in the leader bleb. However, analysis of actin density in the cortical band of the cell body showed that over-expression of Emerald-mEps8Δbund had no effect compared to over-expression of Emerald-mEps8 alone, and caused no detectable changes in the organization of MII-RLC (*Figure 4F*, *Figure 4—figure supplement 1*). AFM analysis showed that while over-expression of Emerald-mEps8 had no effect on cortical tension and intracellular pressure (*Figure 4H–I* and *Supplementary file 1*), over-expression of Emerald-mEps8Δbund significantly reduced cortex tension and intracellular pressure compared to untreated controls (*Figure 4H–I* and *Supplementary file 1*). These results show that the actin bundling activity of Eps8 facilitates the formation of large leader blebs by promoting cortical tension and intracellular pressure.

To further test the notion that F-actin bundling is critical to leader bleb formation and cell mechanical properties, we sought to determine if expression of a different actin bundling protein affected leader blebs or could rescue the defect in leader bleb size induced by loss of Eps8. We over-expressed the actin bundling protein α-actinin (*Podlubnaya et al., 1975*) tagged with EGFP together with FusionRed-F-tractin in either WT or Eps8-KD cells under non-adherent confinement. Confocal imaging of EGFP-α-actinin in either WT or Eps8-KD cells showed a remarkably similar localization as Emerald-mEps8 (*Figure 1I*, *Figure 1—figure supplements 1, 2*), with α-actinin concentrated in leader blebs where it was localized in a gradient along circumferential actin bundles, although the labelling of the bundles was more continuous and less punctate than that of Emerald-mEps8 (*Figure 4—figure supplement 2*). Quantitative analysis showed that in Eps8-KD cells, over-expression of α-actinin was

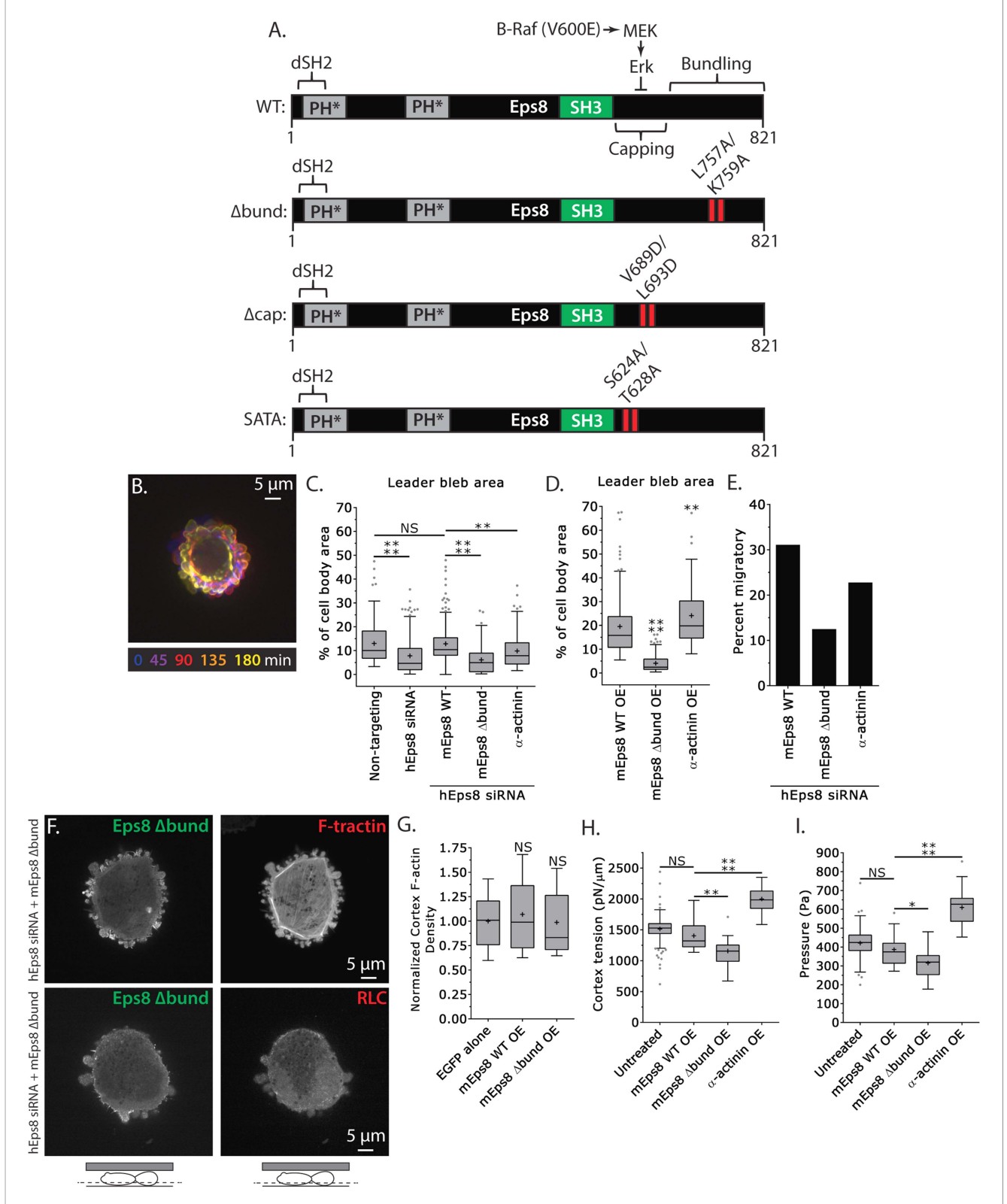

**Figure 4**. Actin bundling by Eps8 promotes cortex tension and intracellular pressure to drive leader bleb formation. (**A**) Schematic representation of wild-type (WT) mouse Eps8 (mEps8) mutant constructs. Top: wild-type Eps8. '*' indicates one part of the split PH domain, 'dSH2' indicates degenerate SH2, 'capping' indicates domain required for actin capping and which is subject to negative regulation by the Raf/MEK/Erk pathway, 'bundling' indicates the domain required for actin bundling. 'Δbund' indicates the bundling defective double point mutant L757A/K759A (red bars), 'Δcap' indicates the capping defective double point mutant V689D/L693D (red bars), 'SATA' indicates double alanine point mutation of Erk phosphorylation sites S624A/T628A (red

*Figure 4. continued on next page*

*Figure 4. Continued*

bars). (**B**–**I**) Images and analyses of A375 cells plated on glass and confined under an agar slab. (**C**, **D**, **G**–**I**) Tukey box plots in which '+' and line denote the mean and median, respectively. (**C**, **D**) Quantification of leader bleb area expressed as a % of cell body area for (**C**) cells treated with non-targeting or Eps8 siRNAs, with or without the additional expression of Emerald-mEps8, Emerald-mEps8Δbund or EGFP human α-actinin or (**D**) cells over-expressing (OE) Emerald-mEps8, Emerald-mEps8Δbund or EGFP human α-actinin. Data for mEps8 are re-displayed from *Figure 2* for comparison. (**E**) Quantitation of the percent of cells that migrate from time-lapse phase contrast videos, treatments as in (**C**). (**F**) Confocal images through the ventral Z-plane of a cell depleted of Eps8 and co-expressing Emerald-mEps8Δbund and either FusionRed-F-tractin (top) or FusionRed-myosin II regulatory light chain (bottom, RLC). (**G**) Analysis of cortical actin density (see Materials and methods) in the cell body from images of phalloidin, treatments as in (**D**), normalized to the mean value of over-expression of soluble EGFP (EGFP alone). (**H**, **I**) Cortex tension (**H**) and intracellular pressure (**H**) determined from AFM analysis of cells under the conditions described in (**D**). *p ≤ 0.05, **p ≤ 0.01, ***p ≤ 0.001, ****p ≤ 0.0001, NS: p > 0.05. See also *Video 4*.

The following figure supplements are available for figure 4:

**Figure supplement 1**. Eps8 bundling activity is not required for myosin II localization to the cortex of A375 cells.

**Figure supplement 2**. Ectopically expressed α-actinin localizes to the leader bleb cortex.

sufficient to partially rescue their defect in leader bleb size, but not to the same extent as expression of Emerald-mEps8Δbund (*Figure 4C–D* and *Supplementary file 1*). In addition, α-actinin over-expression in Eps8-KD cells partially rescued the effects of loss of Eps8 in cell migration (*Figure 4E*). AFM analysis showed that over-expression of α-actinin increased cortical tension and intracellular pressure by more than 50% compared to either untreated cells or cells over-expressing Emerald-mEps8 (*Figure 4H–I* and *Supplementary file 1*). These results demonstrate the critical role of actin bundling in promoting cortical tension and generating intracellular pressure to mediate the formation of leader blebs, however they show that other functions or regulation of Eps8 cannot be compensated by α-actinin.

## Eps8 actin capping activity limits leader bleb size by decreasing actin density and mechanical properties in the cell body cortex, and antagonizing actin bundling at the leader bleb tip

We next sought to determine whether the actin filament capping activity of Eps8 is required for leader bleb formation, cytoskeletal organization, and cellular mechanical properties. We made aspartic acid substitutions within the C-terminus of Emerald-tagged mouse Eps8 (V689D/L693D, referred to as Emerald-mEps8Δcap) which have been previously shown to specifically block Eps8 capping activity (*Figure 4A*) (*Hertzog et al., 2010*) and regulate blebbing during cell division (*Werner et al., 2013*). Strikingly, when confined under agar, Eps8-KD cells expressing Emerald-mEps8Δcap formed significantly larger leader blebs and the proportion of cells migrating nearly doubled compared to either untransfected controls or to Eps8-KD cells expressing Emerald-mEps8 (*Figure 5A,B,D*, *Supplementary file 1* and *Video 5*). Over-expression of Emerald-mEps8Δcap similarly increased leader bleb area compared to over-expression of Emerald-mEps8 (*Figure 5C* and *Supplementary file 1B*). Confocal imaging of confined Eps8-KD cells and analysis of re-expressed protein distribution together with FusionRed-F-tractin or -MII-RLC showed that unlike the gradient of Emerald-mEps8 with the

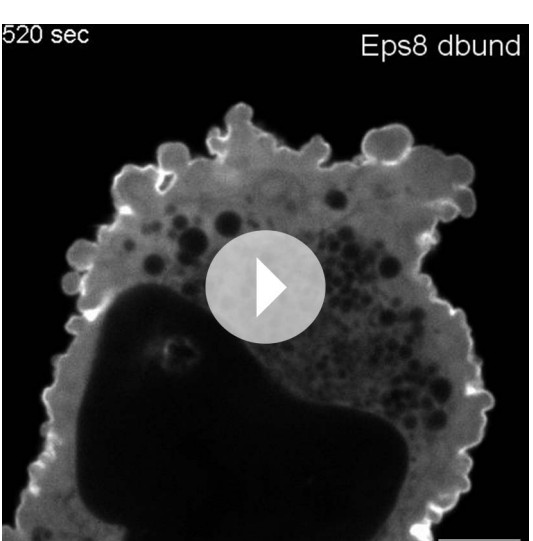

**Video 4.** Formation of large leader blebs requires actin bundling by Eps8. Central Z-plane confocal time-lapse video showing Emerald-Eps8 dbund (L757A/K759A mutations in mouse Eps8 that block its bundling activity) dynamics and small blebs in an A375 cell that has been depleted of Eps8 by siRNA and confined between uncoated glass and an agar pad. Scale bar: 5 μm, elapsed time in seconds shown.

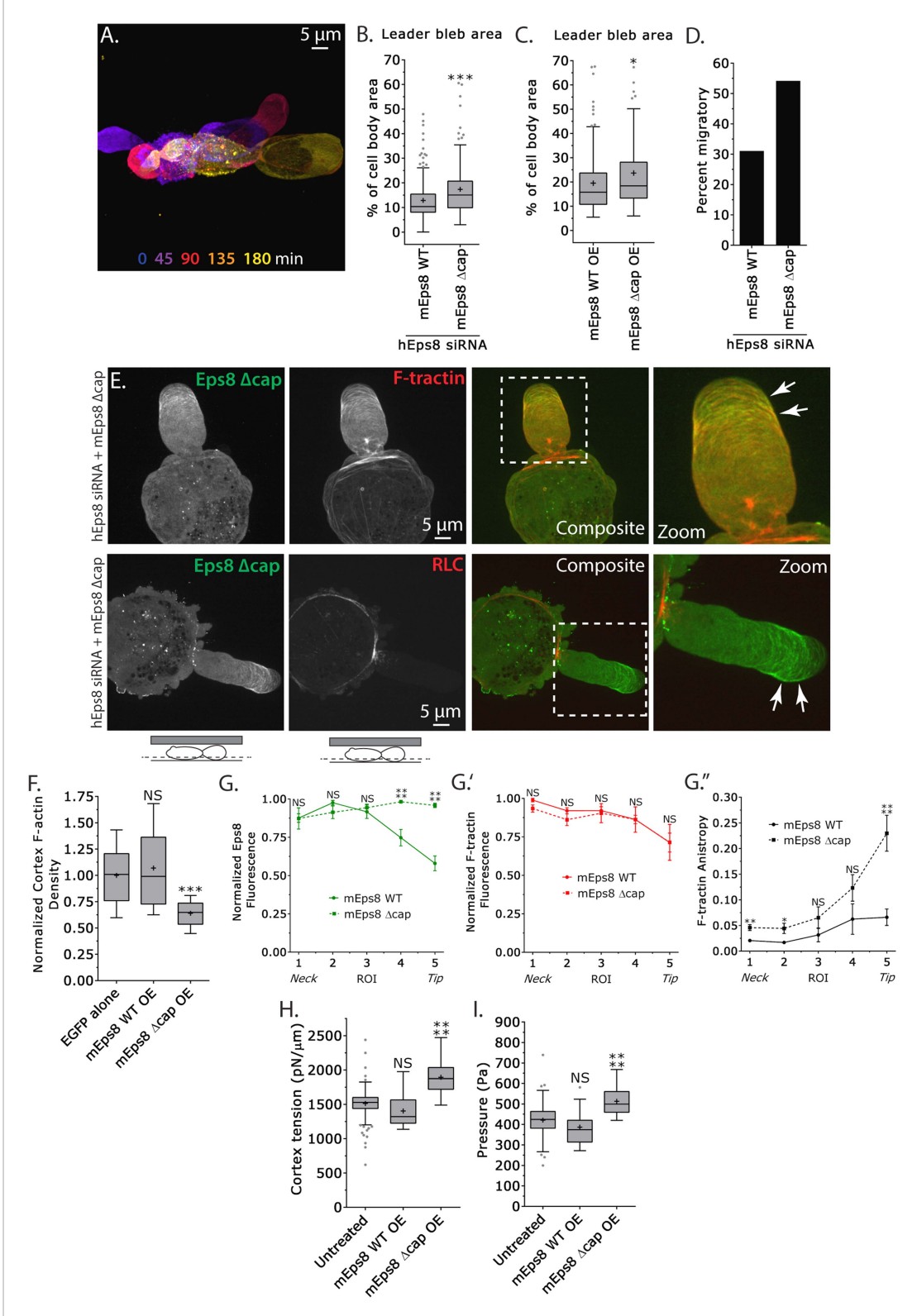

**Figure 5.** Eps8 actin capping activity limits leader bleb size by decreasing actin density and mechanical properties in the cell body cortex, and antagonizing actin bundling at the leader bleb tip. (A–I) Images and analyses of A375 cells plated on glass and confined under an agar slab. (A) Color encoded time-overlay of confocal images through the central Z-plane of a cell depleted of Eps8 by siRNAs targeting human Eps8 (hEps8 siRNA) and expressing Emerald-mEps8Δcap (see *Figure 4A*). (B, C, F, H, I) Tukey box plots in which '+' and line denote the mean and median, respectively. (B, C) Quantification of leader bleb area expressed as a % of cell body area for (B) cells treated with hEps8 siRNAs and additionally expressing wild type

*Figure 5. continued on next page*

Figure 5. Continued

(WT) Emerald-mEps8 or Emerald-mEps8Δcap or (**C**) cells over-expressing (OE) Emerald-mEps8-WT or Emerald-mEps8Δcap. Data for mEps8 WT are re-displayed from *Figure 2* for comparison. (**D**) Quantitation of the percent of cells that migrate from time-lapse phase contrast videos, treatments as in (**B**). (**E**) Confocal images through the ventral Z-plane of cells depleted of Eps8 and co-expressing Emerald-mEps8Δcap and either FusionRed-F-tractin (top) or FusionRed-myosin II regulatory light chain (bottom, RLC). (**F**) Analysis of cortical actin density (see Materials and methods) in the cell body from images of phalloidin, treatments as in (**C**), normalized to the mean value of over-expression of soluble EGFP (EGFP alone). (**G**) Regional analysis of the average fluorescence intensity (**G, G′**, normalized to maximum) Emerald-mEps8 or Emerald-mEps8Δcap (**G**) or FusionRed-F-tractin (**G′**) and bundle anisotropy (**G″**) of FusionRed-F-tractin along leader blebs in cells treated with hEps8 siRNA. Each point represents the average value in a region of interest (ROI) that is 20% of the length of the leader bleb. (**H, I**) Cortex tension (**H**) and intracellular pressure (**H**) determined from AFM analyses of cells under the conditions described in (**D**). *p ≤ 0.05, **p ≤ 0.01, ***p ≤ 0.001, ****p ≤ 0.0001, NS: p > 0.05. See also *Video 5*.

The following figure supplement is available for figure 5:

**Figure supplement 1**. Eps8 capping activity is not required for myosin II localization to the cortex or leader bleb in A375 cells.

highest level at the leader bleb base, Emerald-mEps8Δcap was equally distributed along the length of the leader bleb or slightly concentrated at the tip (*Figure 5E,G*). Analysis of actomyosin organization showed that compared to expression of Emerald-mEps8, expression of Emerald-mEps8Δcap in Eps8-KD cells decreased the level of actin in the dense cortical band of the cell body, although the density gradient of actin along the leader bleb and the distribution of MII-RLC was unchanged (*Figure 5E,F,G′* and *Figure 5—figure supplement 1*). In contrast, the actin bundle anisotropy gradient in the leader bleb was highly enhanced, with Eps8-KD cells expressing Emerald-mEps8Δcap exhibiting nearly fivefold higher actin bundle anisotropy near the tips of their leader blebs compared to Eps8-KD cells expressing Emerald-mEps8 (*Figure 5G″*). AFM analysis of cortex mechanics showed that compared to over-expression of Emerald-mEps8, over-expression of Emerald-mEps8Δcap significantly increased cortex tension and intracellular pressure (*Figure 5H–I* and *Supplementary file 1*). Together, these results show that the capping activity of Eps8 decreases actin density and mechanical properties of the cortex in the cell body, but acts to antagonize actin bundle formation in the distal region of leader blebs, and together this limits leader bleb size. This further suggests that the capping activity of Eps8 may be regionally regulated to maintain a gradient of actin bundle organization in the leader bleb.

## Erk activity spatially regulates actin organization to mediate leader bleb-based migration

Erk-dependent phosphorylation of Eps8 on S624 and T628 has been shown to inhibit actin capping by Eps8 without altering its filament binding activity (*Menna et al., 2009*), and our above results suggest that capping may be regionally regulated in melanoma cells migrating under non-adhesive confinement. Because A375 cells are known to have hyperactivation of the Erk pathway, we sought to test whether Erk activity or phospho-regulation on S624 and T628 of Eps8 regulates leader bleb formation, actin organization, and cortex mechanics. We first exploited the highly specific inhibitor U0126 to block Erk activity (*Figure 6A*). We validated that U0126 inhibited Eps8 phosphorylation in A375 cells by expressing either Emerald-mEps8 or EGFP as

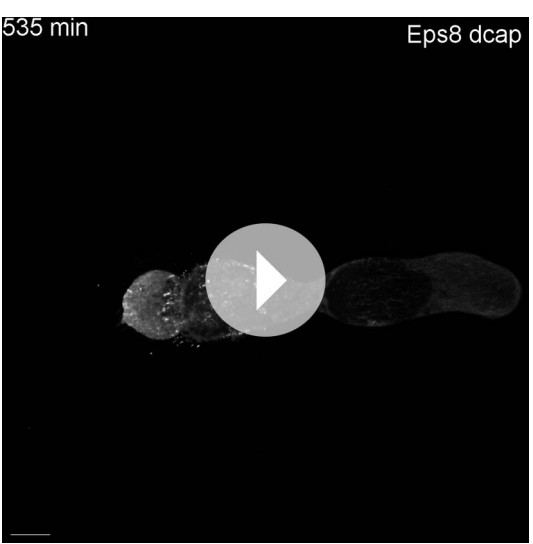

**Video 5.** Actin capping by Eps8 limits leader bleb size. Ventral Z-plane confocal time-lapse video showing Emerald-Eps8 dcap (V689D/L693D mutations in mouse Eps8 that block its actin capping activity) dynamics and large leader bleb formation in an A375 cell that has been depleted of Eps8 by siRNA and confined between uncoated glass and an agar pad. Scale bar: 5 μm, elapsed time in minutes shown.

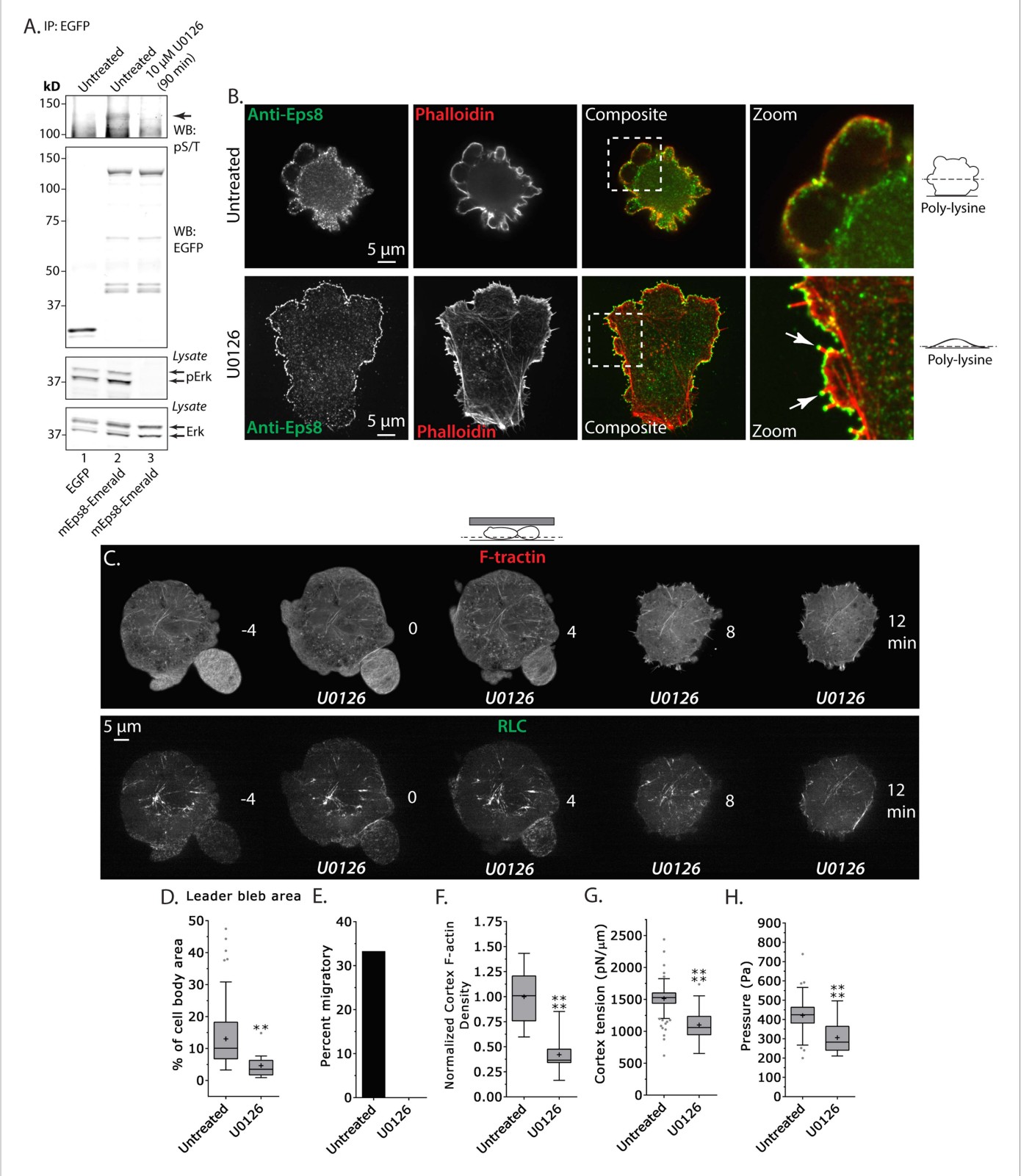

**Figure 6**. Inhibition of Erk activity perturbs leader bleb-based migration. (**A**) Top two panels: Western blot analysis of anti-GFP immunoprecipitates from lysates of A375 cells that were expressing EGFP or Emerald-mEps8 that were or were not treated with 10 μM U0126 to inhibit Erk. Blot was probed with antibodies specific to phospho-serine and phospho-threonine (pS/T, upper panel) or GFP (upper middle panel). Bottom two panels: Western blot analysis of Erk and Erk phosphorylated on T202/Y204 (pErk) in lysates of untreated A375 cells that were expressing EGFP or Emerald-mEps8 that were or were not

*Figure 6. continued on next page*

*Figure 6. Continued*

treated with 10 µM U0126. (**B**) Immuno-localization of endogenous Eps8 (green) and phalloidin staining of actin (red) in untreated and U0126 (10 µM)-treated A375 cells that were non-specifically adhered to poly-L-Lysine coated glass. Arrows: Eps8 localizing to the tips of filopodia. (**C–H**) Images and analysis of A375 cells plated on glass and confined under agarose. (**C**) Time-lapse confocal image series of a cell that was co-expressing FusionRed-F-tractin (top) and EGFP tagged myosin II regulatory light chain (RLC, bottom). Time indicates minutes relative to perfusion with 10 µM U0126. (**D, F, G, H**) Tukey box plots in which '+' and line denote the mean and median, respectively, treatments as in (**A**). (**D**) Quantification of leader bleb area expressed as a % of cell body area. (**E**) Quantitation of the percent of cells that migrate from time-lapse phase contrast videos. (**F**) Analysis of cortical actin density (see Materials and methods) in the cell body from images of phalloidin, normalized to the mean value of untreated cells. (**G, H**) Cortex tension (**G**) and intracellular pressure (**H**) determined from AFM analyses of cells. *p ≤ 0.05, **p ≤ 0.01, ***p ≤ 0.001, ****p ≤ 0.0001, NS: p > 0.05. See also *Video 6*.

The following figure supplement is available for figure 6:

**Figure supplement 1**. Erk inhibition causes A375 cell flattening.

a control in cells treated with or without 10 µM U0126 for 90 min, performing immunoprecipitation of the expressed proteins with anti-GFP antibodies from cell lysates, followed by western blot analysis with antibodies specific to Erk, activated Erk phosphorylated on T202/Y204, or phospho-serine and phospho-threonine (pS/T). This analysis showed, as expected, that U0126 inhibited Erk activation independent of the expression of Eps8 constructs. Similarly, U0126 strongly reduced the pS/T level in immunoprecipitated Emerald-mEps8 compared to untreated cells. This indicates that U0126 blocks Erk-mediated Eps8 phosphorylation.

We then examined the role of Erk activity in cytoskeletal organization and Eps8 localization. We localized endogenous Eps8 and actin in cells plated on poly-L-Lysine to mediate non-specific adhesion during immunostaining. This showed that U0126 treatment caused cells to flatten out at their base where they attached to the coverslip and to form actin bundles in the cell center and lamellipodia and filopodia with Eps8 on their tips, although they still possessed small blebs containing Eps8 on their dorsal surface (*Figure 6B* and *Figure 6—figure supplement 1*). When cells co-expressing FusionRed-F-tractin and EGFP-MII-RLC were confined under non-adherent conditions and perfused with U0126, this remarkably induced rapid retraction of leader blebs and formation of actin bundles in the center of the cell body that lacked myosin II (*Figure 6C* and *Video 6*). Quantification showed that compared to untreated control, U0126 significantly reduced leader bleb area and density of the cortical actin band in the cell body, and completely blocked leader-bleb based migration (*Figure 6D,E,F*). AFM analysis of cortical mechanics showed that U0126 reduced both cortical tension and intracellular pressure compared to control (*Figure 6G–H* and *Supplementary file 1*). These results illustrate two important points. First, they show that Erk activity mediates spatial regulation of the actin cytoskeleton, such that it promotes actin density in the cortex and inhibits actin bundling in the cell center. Second, Erk activity is required for maintaining cortical tension and intracellular pressure to drive leader bleb formation for adhesion-independent migration under confinement.

## MEK/Erk-mediated phosphorylation of S624 and T628 coordinates Eps8 capping and bundling activities to mediate leader bleb-based migration

We next sought to test specifically if phospho-regulation on the Erk sites (S624 and T628) of Eps8 were critical to leader bleb formation and cytoskeletal regulation. We generated a non-phosphorylatable mutant of Emerald-mEps8 (*Figure 4A*, S624A/T628A, referred to as Emerald-mEps8-SATA) that has been shown to have constitutive capping activity in vitro and inhibit filopodia formation in neurons (*Menna et al., 2009*). We expressed Emerald-mEps8 or Emerald-mEps8-SATA in either control or Eps8-KD cells and subjected them to confinement under non-adhesive conditions. This showed that either reconstitution of Eps8-KD or over-expression with Emerald-mEps8-SATA blocked leader bleb formation, significantly reducing leader bleb size by ~50% and strongly inhibiting cell migration compared to expression of Emerald-mEps8 in either wild type or Eps8-KD cells (*Figure 7A–D*, *Supplementary file 1* and *Video 7*). Thus, S624 and T628 in Eps8 are required for leader bleb formation and migration under non-adhesive confinement. This further suggests that down-regulation of capping activity by phosphorylation at these sites promotes leader blebs, in agreement with our finding that the capping-deficient Eps8Δcap enhances leader blebs.

We then examined the effects of the Eps8 phospho-mutant on cytoskeletal organization. Interestingly, expression of Emerald-mEps8-SATA together with FusionRed-F-tractin showed that the

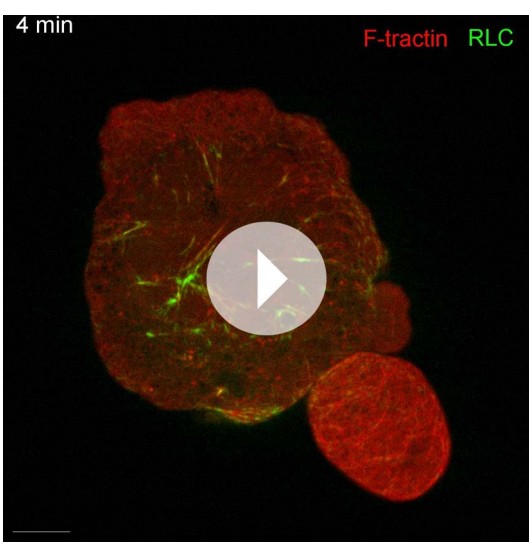

**Video 6.** MEK/Erk activity regulates the organization of the actin cortex and is required to form large leader blebs. Ventral Z-plane confocal time-lapse video showing actin (FusionRed-F-tractin, red) and myosin II (EGFP-myosin II regulatory light chain, RLC, green) dynamics and leader bleb retraction in an A375 cell confined between uncoated glass and an agar pad. When the word 'U0126' appears, 10 μM U0126 was perfused into the media to inhibit MEK/Erk. Scale bar: 5 μm, elapsed time in minutes shown.

Eps8 phospho-mutant was depleted from the cortex and instead formed thick bundle structures in the cell center (*Figure 7E*), similar to the actin bundles seen in cells treated with U0126 (*Figure 6C*), but the bundles were not labeled with F-tractin (*Figure 7E*). However, fixation and staining with antibodies to vimentin or with fluorescent phalloidin showed that the Emerald-mEps8-SATA cables did not co-localize with intermediate filaments, but were dense with actin, suggesting that F-tractin and Eps8 may compete for the same binding site in actin bundles (*Figure 7—figure supplement 1A,B*). Analysis of fluorescent phalloidin intensity in the cortical band of the cell body showed that Emerald-mEps8-SATA did not alter actin density compared to Emerald-mEps8 (*Figure 7F*). In addition, co-expression of Emerald-mEps8-SATA together with FusionRed MII-RLC in Eps8-KD cells showed that myosin II continued to localize to the peripheral cortex, but the central Eps8/actin bundles lacked myosin II (*Figure 7E*). Quantification of the percentage of cells with Eps8/actin bundles showed that expression of F-tractin did not induce bundle formation, while over-expression of Emerald-mEps8 or Emerald-mEps8-SATA induced bundles in ~30% or ~70% of cells, respectively (*Figure 7H*). This indicates that Emerald-mEps8-SATA promotes the formation of excessive non-contractile actin bundles in the cell center.

Together with our previous results, this suggests that the S624 and T628 Erk phosphorylation sites in Eps8, are required for suppressing central actin bundles, while other targets of MEK/Erk regulate cortical actin density.

We then sought to determine how the S624A/T628A mutations in Eps8 mediated their effects on the cytoskeleton. We first tested whether this was due to a lack of Erk-mediated regulation on these sites. Treatment of non-adherent cells expressing either Emerald-mEps8 or Emerald-mEps8-SATA with 10 μM U0126 for 90 min followed by fixation and phalloidin staining showed that Erk inhibition induced Eps8/actin cables to the same extent, independent of the S624A/T628A mutations (*Figure 7G,H*). Because phosphorylation of S624 and T628 are known to inhibit the actin capping activity of Eps8, and yet non-phosphorylatable mutation of these sites (which would be expected to have constitutive capping) produced an ectopic actin cable effect, we wondered if the formation of Eps8/actin cables induced by Erk inhibition required either the capping or bundling activities of Eps8. Over-expression of either Emerald-mEps8Δcap or Emerald-mEps8Δbund in either the presence or absence of U0126 did not induce Eps8/actin cables, indicating that both capping and bundling activities are required for cable formation induced by Erk inhibition (*Figure 7G,H*). Furthermore, over-expression of Emerald-mEps8Δcap in cells treated with U0126 and confined under agar could not rescue the loss of leader blebs induced by perfusion of Erk inhibitor (*Figure 7—figure supplement 2*), indicating that Erk promotes leader blebs by other mechanisms in addition to down-regulation of Eps8 capping activity. Finally, we found that over-expression of an S624E/T628E Eps8 mutant (*Menna et al., 2009*) could not recapitulate the effects of Emerald-mEps8Δcap in cells, suggesting that the E substitutions do not act as phospho-mimics to inhibit Eps8 capping activity in cells (not shown). However, together our results show that MEK/Erk-mediated phosphorylation of Eps8 on S624 and T628 inhibits the bundling- and capping-dependent formation of central actin cables by Eps8. This also suggests that in addition to directly regulating actin capping (*Menna et al., 2009*), these sites may affect the bundling activity of Eps8 in cells by an indirect mechanism.

We then tested the role of S624 and T628 of Eps8 and their regulation by Erk in cortex mechanics. AFM analysis showed that compared to untreated or cells expressing Emerald-mEps8,

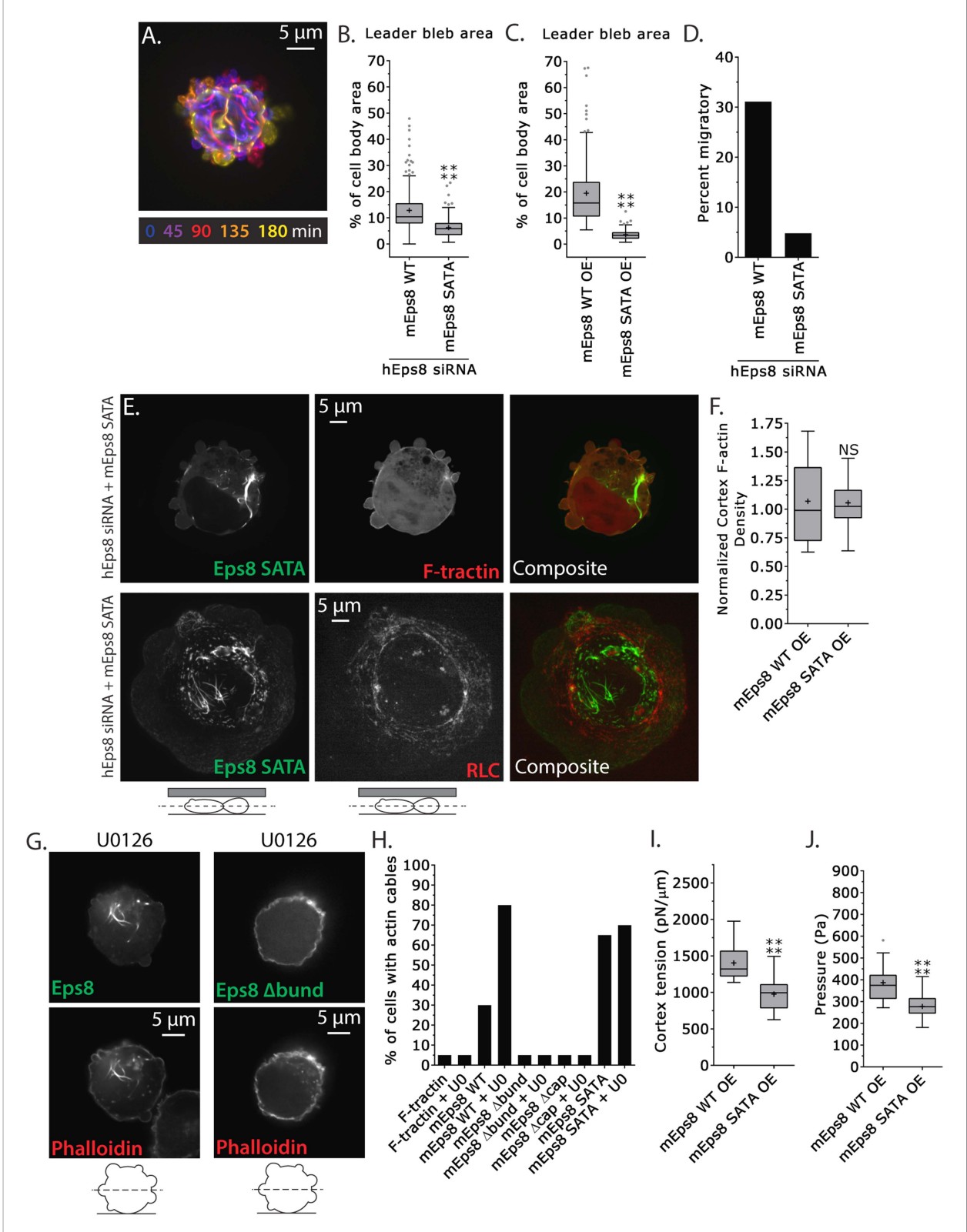

**Figure 7**. MEK/Erk-mediated phosphorylation of S624 and T628 coordinates Eps8 capping and bundling activities to mediate leader bleb-based migration. (A–J) Images and analyses of A375 cells plated on glass and confined under an agar slab. (A) Color encoded time-overlay of confocal images through the central Z-plane of a cell depleted of Eps8 by siRNAs targeting human Eps8 (hEps8 siRNA) and expressing Emerald-mEps8 bearing double alanine point mutation of Erk phosphorylation sites S624A/T628A (Emerald-mEps8-SATA). (B, C, F, I, J) Tukey box plots in which '+' and line denote the

**Figure 7. Continued**

mean and median, respectively. (**B–C**) Quantification of leader bleb area expressed as a % of cell body area for (**B**) cells treated with hEps8 siRNAs and additionally expressing wild type (WT) Emerald-mEps8 or Emerald-mEps8-SATA or (**C**) cells over-expressing (OE) Emerald-mEps8-WT or Emerald-mEps8-SATA. (**D**) Quantitation of the percent of cells that migrate from time-lapse phase contrast videos, treatments as in (**B**). (**E**) Confocal images through the central Z-plane of cells depleted of Eps8 and co-expressing Emerald-mEps8-SATA and either FusionRed-F-tractin (top) or FusionRed-myosin II regulatory light chain (bottom, RLC). (**F**) Analysis of cortical actin density (see Materials and methods) in the cell body from images of phalloidin, treatments as in (**C**), normalized to the mean value from over-expression of Emerald-mEps8 WT. (**G**) Confocal images through the central Z-plane of cells treated with 10 μM U0126 and over-expressing either Emerald-mEps8 WT or Emerald-mEps8Δbund (see *Figure 4A*) and actin stained with fluorescent phalloidin. (**H**) Quantitation of the percent of cells containing central actin bundles from confocal images of fluorescent phalloidin-stained cells. 'F-tractin' indicates over-expression of FusionRed-F-tractin, 'UO' indicates treatment with 10 μM U0126, 'mEps8,' 'mEps8Δcap,' 'mEps8Δbund,' 'mEps8-SATA,' indicate over-expression of EGFP-tagged versions of the respective constructs (see *Figure 4A*). (**I, J**) Cortex tension (**I**) and intracellular pressure (**J**) determined from AFM analyses of cells under the conditions described in (**C**). *p ≤ 0.05, **p ≤ 0.01, ***p ≤ 0.001, ****p ≤ 0.0001, NS: p > 0.05. See also *Video 7*.

The following figure supplements are available for figure 7:

**Figure supplement 1**. Non-phosphorylatable Eps8 co-localizes with F-actin and not intermediate filaments.

**Figure supplement 2**. Inhibition of Eps8 capping activity is not sufficient to maintain leader blebs in the presence of Erk inhibitor.

Emerald-mEps8-SATA expression reduced cortex tension and intracellular pressure to a level similar to treatment with U0126 (*Figures 7I,J, 6G,H* and *Supplementary file 1*). This was surprising, considering that this mutant had no effect on cortical actin density. However, it is possible that the cortical mechanics could be altered by the redistribution of mutant Eps8 from the cortex to central actin bundles. Together, these results suggest that Erk-mediated phosphorylation of S624 and T628 coordinates the local co-regulation of Eps8 capping and bundling activities to control cortical tension and intracellular pressure to mediate leader bleb formation and migration of confined cells.

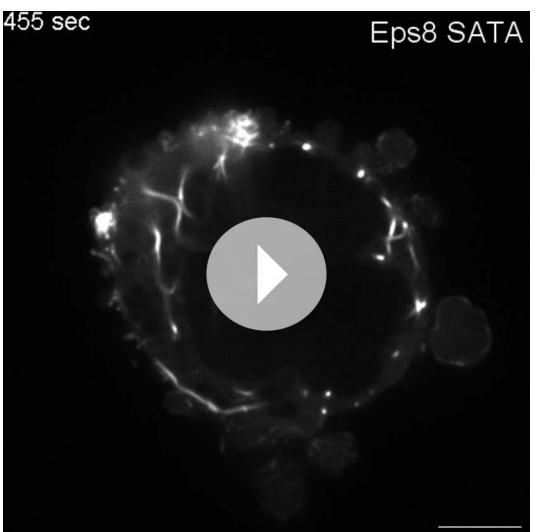

**Video 7.** Erk phosphorylation of Eps8 is necessary for large leader bleb formation. Ventral Z-plane confocal time-lapse video showing Emerald-Eps8 SATA (S624A/T628A mutations in mouse Eps8 that block its ability to be phosphorylated by Erk) dynamics and small blebs in an A375 cell that has been depleted of Eps8 by siRNA and confined between uncoated glass and an agar pad. Scale bar: 5 μm, elapsed time in seconds shown.

## Erk activity is concentrated in a gradient across leader blebs and its mobility is restricted by a diffusion barrier at the bleb neck

Our demonstrations that Erk activity and regional regulation of Eps8 activity are essential to leader bleb formation suggests that Erk activity itself may be regionally regulated in migrating melanoma cells. Because immunofluorescense (of phospho-Eps8 or active Erk) is not possible in cells confined under agar, we turned to a Fluorescence Resonance Energy Transfer (FRET)-based biosensor of active Erk called 'Erk Kinase Activity Reporter containing EV linker' (EKAREV) (*Harvey et al., 2008*; *Komatsu et al., 2011*). In short, a CFP-tagged phospho-peptide-binding domain is connected by a FRET-optimized 'EV linker' to a YFP-tagged Erk substrate peptide, and FRET is obtained when the substrate is phosphorylated by Erk. We first used the reporter to localize Erk activity in cells adhered to fibronectin-coated glass. Confocal imaging showed that EKAREV (CFP channel) was soluble, excluded from membranous organelles, and diffusely localized throughout the cell, while YFP/CFP ratio imaging revealed a low FRET signal indicating low Erk activity throughout the cell (*Figure 8A*). In contrast, in non-adherent,

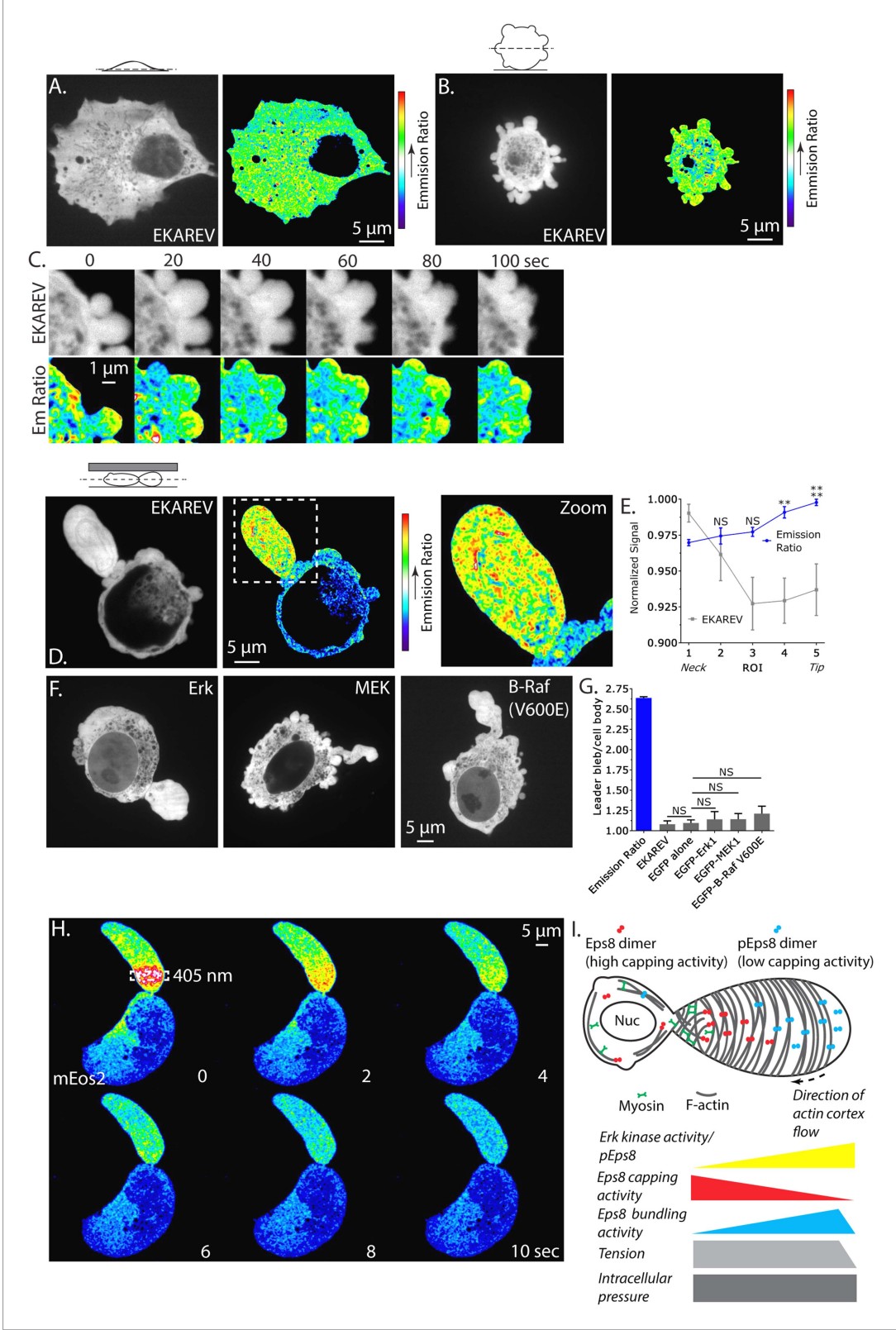

**Figure 8**. Erk activity is concentrated in a gradient across leader blebs by a diffusion barrier at the bleb neck. (**A–E**, **G**) A375 cells expressing the EKAREV biosensor in which a CFP-tagged phospho-peptide-binding domain is connected by the EV linker to a YFP-tagged Erk substrate peptide and FRET is obtained when the substrate is phosphorylated by Erk. (**A**) (Left) Confocal image at the ventral Z-plane of the distribution of EKAREV (CFP channel)
*Figure 8. continued on next page*

*Figure 8. Continued*

in a cell plated on fibronectin-coated glass. (Right) Pseudocolored ratio image of EKAREV FRET (YFP over CFP emission). (**B**) (Left) Confocal image at the central Z-plane of the distribution of EKAREV (CFP channel) in a cell plated on glass. (Right) Pseudocolored ratio image of EKAREV FRET. (**C**) (Top) Time-lapse confocal image series at the central Z-plane of the distribution of EKAREV (CFP channel) in a cell plated on glass. (Bottom) Pseudocolored ratio image of EKAREV FRET. (**D–H**) Images and analysis of A375 cells cell plated on glass and overlaid with an agar slab. (**D**) (Left) Confocal image at the central Z-plane of the distribution of EKAREV (CFP channel). (Center) Pseudocolored ratio image of EKAREV FRET. (Right) Zoom of the boxed area. (**E**) Regional analysis of the average fluorescence intensity of EKAREV (CFP channel) or the EKAREV FRET (YFP over CFP emission) (normalized to maximum) along leader blebs. Each point represents the average value in a region of interest (ROI) that is 20% of the length of the leader bleb. (**F**) Confocal image at the central Z-plane of the distribution of EGFP tagged Erk, MEK or B-Raf (V600E) in cells that are confined under agarose. (**G**) Quantification of the average ratio of signal in the leader bleb to that in the cell body for (**D, F**). 'Emission ratio' indicates EKAREV FRET signal (YFP over CFP emission), 'EKAREV' indicates EKAREV CFP channel, 'EGFP alone' indicates soluble EGFP. (**H**) A375 cell expressing soluble mEos2. Time-lapse confocal image series at the central Z-plane. Box indicates the region near the neck of the leader bleb to which a pulse of 405 nm light was applied to locally photo-convert mEos2 from green to red fluorescence. Pseudocolor indicates the magnitude of red fluorescence from mEos2 after photo-conversion. (**I**) Speculative model for Eps8 function during leader bleb-based migration. $*p \leq 0.05$, $**p \leq 0.01$, $***p \leq 0.001$, $****p \leq 0.0001$, NS: $p > 0.05$. See also *Videos 8, 9*.

The following figure supplement is available for figure 8:

**Figure supplement 1**. The Erk inhibitor U0126 reduces EKAREV FRET and A375 cell blebbing.

blebbing cells, although the EKAREV reporter was evenly distributed, the ratio image showed heightened levels of FRET indicating higher Erk activity specifically in blebs (*Figure 8B,C*). Importantly, treatment with U0126 reduced the EKAREV FRET ratio to minimal levels (*Figure 8—figure supplement 1*). Remarkably, time-lapse ratio imaging at 5 s intervals showed that a flash of high FRET appeared in the bleb periphery just after protrusion, and was maintained until bleb retraction (*Figure 8C*). Thus, EKAREV reveals spatially and temporally localized Erk activity in blebs of non-adherent cells.

We next sought to localize Erk kinase activity in non-adherent cells confined under agarose. Strikingly, confocal YFP/CFP ratio imaging revealed a strong concentration of high FRET signal indicating high Erk activity within leader blebs compared to the cell body (*Figure 8D* and *Video 8*). Quantification of the magnitude of FRET signal in the leader bleb relative to the cell body showed >250% enrichment of Erk activity in the leader bleb (*Figure 8G*). Furthermore, regional analysis of the magnitude of FRET signal distribution within leader blebs showed a shallow gradient, with Erk activity highest at the distal tip (*Figure 8E*). To determine if this effect could be caused by an enrichment of signaling proteins within leader blebs, we localized either soluble EGFP, or EGFP tagged versions of Erk, MEK or activated B-Raf (V600E) in cells confined under agarose (*Figure 8F*). Quantification showed that EGFP alone as well as EKAREV were enriched by ∼10–12% in the leader bleb relative to the cell body (*Figure 8G*). Although Erk, MEK and B-Raf (V600E) were all enriched within leader blebs by 15–20% relative to the cell body (*Figure 8F,G*), this was not significantly higher than the enrichment of EGFP alone. Therefore, slight enrichment of soluble proteins within leader blebs may be a general phenomenon, likely because membranous organelles that exclude cytoplasmic markers are largely excluded from the leader

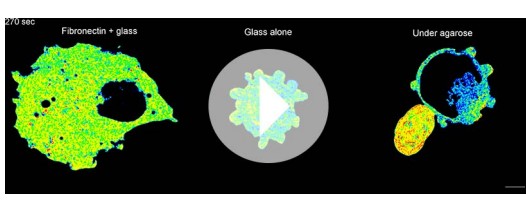

**Video 8.** The EKAREV biosensor reveals concentrated Erk kinase activity within large leader blebs. Comparison of central Z-plane confocal time-lapse videos of Erk kinase activity in A375 cells plated on human plasma fibronectin coated glass (left), uncoated glass (middle) and confined between uncoated glass and an agar pad (right). Cells were expressing the EKAREV Erk activity biosensor. Pseudocolor reflects the magnitude of YFP/CFP ratio FRET signal induced by Erk-mediated phosphorylation of the biosensor (red = high, blue = low). Scale bar: 5 µm, elapsed time in seconds shown.

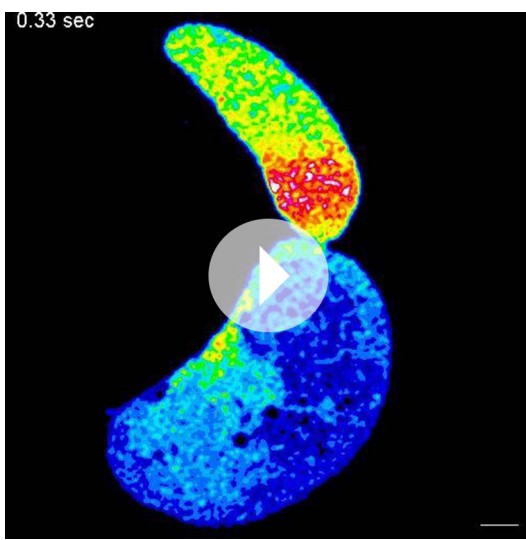

0.33 sec

**Video 9.** The bleb neck restricts the diffusion of photo-converted mEos2 between large leader blebs and the cell body. Central Z-plane confocal time-lapse video of freely diffusing mEos2 before and after activation with a 0.5 s pulse of 405 nm laser light within a defined region (box) of a leader bleb showing slow diffusion between a large leader bleb and the cell body. Pseudocolor reflects the magnitude of red fluorescence from mEos2 after photo-conversion (red = high, blue = low). Scale bar: 5 µm, elapsed time in seconds shown.

bleb and maintained in the cell body. Together, our results show that in confined non-adherent cells, Erk activity is highly concentrated in leader blebs where it forms a gradient with peak activity at the bleb tip.

Our observation that membranous organelles were excluded and soluble proteins were slightly concentrated in leader blebs led us to hypothesize that leader blebs trap their contents with a diffusion barrier formed by the constriction between the bleb and cell body. To test this, we utilized soluble monomeric Eos which converts from green to red fluorescence upon exposure to UV light. Using a localized pulse of 405 nm light, we photo-converted mEos in a rectangular region within the leader bleb a short distance away from the neck and then imaged the redistribution of red fluorescence by diffusion within the bleb and cell body. This showed that the majority of the red fluorescence was retained within and diffused throughout the leader bleb before equilibrating with the cell body (*Figure 8H* and *Video 9*). These results indicate that the bleb neck slows diffusion of soluble components between the leader bleb and cell body. Together, these results show that Erk activity concentrates in and forms a gradient across leader blebs, and may be trapped there by a diffusion barrier at the bleb neck.

## Discussion

Our study shows for the first time a critical role for Eps8 and its regulation of the actin cytoskeleton in promoting cell cortex tension and intracellular pressure to induce the rapid, adhesion-independent migration of confined cancer cells. We demonstrate that, in contrast to cells adhered to ECM where Eps8 associates with filopodia and lamellipodia, in non-adherent melanoma cells, Eps8 localizes to bleb membranes as actin assembles. A novel, bleb-based mode of migration was recently described by several labs that occurs when non-adherent, highly contractile cells are squeezed in 3D confinement. The pressure induced by such an environment induces cells to undergo a MAT and adopt a characteristic polarized morphology where rapid cell motility is driven by cortical cytoskeletal flow that generates extracellular friction along a large sausage-shaped 'leader bleb' that drags along the cell body (*Bergert et al., 2015*; *Liu et al., 2015*; *Ruprecht et al., 2015*). We show here that in cancer cells, the actin capping and bundling scaffold protein Eps8 is required to promote this morphological transition under confinement by enhancing cortical tension and promoting increased intracellular pressure. Our manipulations of Eps8 level and activities that defined its requirement for leader bleb formation and migration did not perturb the organization or activity of myosin II. This, together with previous work (*Bergert et al., 2015*; *Liu et al., 2015*; *Ruprecht et al., 2015*), indicates that both contractility and proper organization of the cytoskeleton are critical to generating the cortical mechanical properties conducive to leader bleb-based migration under non-adhesive confinement.

Our experimental results suggest a speculative model for how Eps8 mediates leader bleb-based migration (*Figure 8I*). We find that the bundling activity of Eps8 is required for promoting the mesenchymal-to-leader bleb transition in confined, non-adherent melanoma cells. This identifies a critical role for actin bundles in promoting the cortical inhomogeneity and cellular pressure that have been postulated to mediate symmetry breaking to drive this drastic polarized cell shape change (*Ruprecht et al., 2015*). Once the morphology transition occurs, we find that Eps8 and actin both form a gradient within the leader bleb, decreasing with distance from the neck, but absent from the

very tip. Our demonstration the Eps8 localizes to bleb membranes on a similar timescale as ezrin suggests that their interaction (*Zwaenepoel et al., 2012*) may be critical to Eps8 cortical recruitment. Analysis of actin bundle anisotropy shows that actin bundling forms an opposing gradient, with the most organized actin bundles towards the bleb tip. Actin bundles wrapping around the short axis of the bleb are likely responsible for maintaining the sausage/allantoid shape. Our demonstration that inhibition of Eps8 capping activity further enhances actin bundle organization towards the leading bleb tip suggests that Eps8's bundling and capping activities act antagonistically, such that when capping is down-regulated, Eps8 may be a more efficient bundler. Thus, local regulation of Eps8 capping activity could promote polarization of actin organization with opposing gradients in actin density and bundling along the leader bleb. However, by manipulating Erk activity or the Erk phosphorylation sites in Eps8, we demonstrate that Erk-mediated phosphorylation of S624 and T628 not only down-regulates the capping activity of Eps8 as shown previously (*Menna et al., 2009*), but may in fact coordinate the regulation of Eps8 capping and bundling activities. Our use of a FRET biosensor surprisingly shows that Erk activity is massively concentrated in leader blebs, although the mechanism for this concentration is not clear. However, the high activity is likely trapped there by a diffusion barrier at the bleb neck, and within the leader bleb forms a shallow gradient with peak activity at the bleb tip. This suggests that Erk-mediated down-regulation of Eps8 capping activity at the leader bleb tip could drive the formation of opposing gradients in actin bundling and density along the bleb length. This gradient in actin organization, in turn, may be required for spatial organization of cortical tension, which we predict would mirror the gradient in actin bundling, to maintain the allantoid shape and drive cortical flow in the leader bleb. The increased cortical tension at the bleb tip would be opposed at the bleb neck by myosin II contractility, and the depletion of myosin II and actin from the very tip of the leader bleb would provide a site where high intracellular pressure could promote local protrusion of the leading membrane to mediate the leader-bleb-based migration of confined cells. Thus, our results identify a mechanism by which spatial regulation of Eps8 actin regulatory activities by Erk may promote the rapid, unregulated migration of melanoma cells that may be critical to their highly invasive behavior in vivo.

The confinement of cells simulated here by an under agarose assay is thought to occur in animal cells during intraepithelial migration, within the perivascular space and between muscle fibers (*Charras and Sahai, 2014*). Additionally, leader bleb-based migration may be important to migration between tightly packed cells found in solid tumors. Our observation that inhibition of Erk activity is capable of blocking formation of large leader blebs is consistent with the notion that effectors of the pathway, such as Eps8, are important to the migration of confined cancer cells. The Ras/Raf/MEK/Erk pathway is one of the most frequently upregulated pathways in cancer. In melanoma, this pathway is particularly active because of a commonly found activating mutation in B-Raf V600E (*Davies et al., 2002*) that bypasses Ras and the negative feedback that normally restrains MEK/Erk activity (*Logue and Morrison, 2012*). This feature predisposes melanoma cells to blebbing by supporting a Raf/MEK/Erk/MLCK/RLC/myosin contractility cascade. Drugs that target B-Raf V600E (e.g., vemurafenib) have been shown to have remarkable effects in the short-term. Compensatory upregulation of the pathway by cancer cells frequently limits their effectiveness (*Logue and Morrison, 2012*). However, our work shows that other mechanisms for activating Erk, including via oncogenic Ras mutations in lung cancer cells or via serum factors in osteosarcoma cells also lead to Eps8-dependent bleb-based migration. The work described here elucidates the importance of a specific Erk effector, Eps8, in the migration of confined melanoma cells. Therefore, targeting specific effectors as opposed to canonical signaling enzymes may have therapeutic value. Effectors that impact cell architecture through the regulation of the cytoskeleton could be particularly attractive targets especially for prevention of metastasis.

## Materials and methods

### Cell culture and transfection

A375, A549 and U2OS cells were all obtained from American Type Culture Collection (ATCC, Manassas, VA) and all were maintained for up to 15 passages in DMEM supplemented with 10% FBS, GlutaMAX (Life Technologies, Carlsbad, CA), antibiotic-antimycotic (Life Technologies) and 20 mM Hepes pH 7.4. Lipofectamine 2000 (Life Technologies) and RNAiMAX (Life Technologies) were used to transfect plasmids and small interfering RNAs, respectively. Cells were plated on 6-well glass bottom

plates (In Vitro Scientific, Mountain View, CA) either directly, or after coating with either 5 or 50 µg/ml human plasma fibronectin (Millipore, Billerica, MA) or poly-L-Lysine (Millipore), as noted.

## Confinement of cells under agarose

Agarose slabs for cell confinement (*Bergert et al., 2012*) were made by adding 750 mg of ultrapure agarose (Life Technologies) to 50 ml of 20 mM Hepes (pH 7.4), microwaving briefly, and pouring 4 ml into each well of a 6-well glass bottom plate (In Vitro Scientific). After gelation, a hole was punched in the agarose using a 5 ml plastic test tube. Prior to confining cells, 3 ml of media was pipetted into each well and equilibrated with the agarose overnight. Before use, media was thoroughly vacuumed off and 200 microliters of media containing cells was added to the empty hole punch. To get cells under the agarose, a 1 ml pipette tip was placed into the hole punch containing media and cells and slid just under the agarose to gently lift a portion of the agarose, sucking the cells underneath, and the pad was gently set down. The remaining media and cells were then thoroughly vacuumed out of the hole punch. To prevent drying, the plate was sealed using parafilm. Prior to imaging, the plate was brought up to temperature for 1 hr in an incubator.

## Pharmacological treatments

To inhibit the Erk pathway, a working concentration of 10 µM U0126 (Cell Signaling Technology, Beverly, MA) was prepared by diluting a 1000× stock solution in DMSO into complete media and dissolved using a vortex mixer for 30 s before adding to cells. Phospho-Erk (T202/Y204) (#4370) and Erk (#9102) antibodies purchased from Cell Signaling Technology were used to confirm inhibition of MEK by Western blotting of whole-cell lysates. Cells were treated with U0126 for 30 min for AFM assays and 90 min for light microscopy assays. Cells under agarose were treated by applying media containing U0126 directly on top of the agar pad. Blebbistatin (Sigma Aldrich, St. Louis, MO) was used at 50 µM and applied directly to the cells for 5 min before AFM analyses.

## Plasmids

EGFP-tagged mouse Eps8 (GFP-mEps8) and Emerald-mEps8-SATA (S624A/T628A) were the kind gift of Giorgio Scita (University of Milan). Bundling (L757A/K759A, *Δbund*) and capping (V689D/L693D, *Δcap*) defective versions of Emerald-mEps8 were made using Quick Change II XL (Agilent Technologies, Santa Clara, CA) and the following primers:

### Δbund
Forward primer: CGGAGCACAACTCTTTTCTGCCAACGCAGACGAACTGAGGTCTG
Reverse primer: CAGACCTCAGTTCGTCTGCGTTGGCAGAAAAGAGTTGTGCTCCG

### Δcap
Forward primer: GTCCCAGATGGAAGAGGATCAGGATGAGGACTTCCAGAGGCTGACC
Reverse primer: GGTCAGCCTCTGGAAGTCCTCATCCTGATCCTCTTCCATCTGGGAC

Open reading frames of WT, non-phosphorylatable, Δbund and Δcap versions of mEps8 were then PCR amplified and inserted into pENTR/D-TOPO (Life Technologies) before shuttling into pcDNA 6.2/N-EmGFP-DEST (Life Technologies) using LR Clonase (Life Technologies).

mEmerald and Fusion Red-tagged F-tractin, an acting binding peptide derived from the Itpka protein from rat was generated by cloning into a Clontech-style vector containing the advanced EGFP variant Emerald (F64L, S65T, S72A, N149K, M153T, I167T, A206K). The following primers were used to amplify F-tractin from rat Itpka (NM_031045.2) and generate a 13 amino acid linker separating the F-tractin from the fluorescent protein:
Forward primer containing HindIII site: AGC TCA AGC TTA TGG GCA TGG CGC GAC CAC GGG GCG C
Reverse primer containing BamHI site: CCG GTG GAT CCG ATC CAG ATC CGC CGC AGC GCG CTT CGA AGA GCA GGC GCA GCT CC

The resulting PCR product and mEmerald-N1 were digested by the appropriate restriction enzymes, gel purified, and ligated to yield mEmerald-F-tractin-N-13. Upon sequence verification of the vector, mEmerald-F-tractin-N-13, and FusionRed-N1 were digested by the appropriate restriction enzymes, gel purified, and ligated to yield FusionRed-F-tractin-N-13.

EKAREV was kindly provided by Kazuhiro Aoki (Kyoto University) and Jun-ichi Miyazaki (Osaka University), EGFP-B-Raf V600E was generated by the Advanced Technology Research Facility (NCI, Frederick, MD). Ezrin-GFP (Stephen Shaw), EGFP-Erk1 (Rony Seger), MEK1-GFP (Rony Seger) and mEos2 (Michael Davidson) were obtained from Addgene (Cambridge, MA). FusionRed tagged myosin II regulatory light chain (RLC) and vimentin were purchased from Evrogen (Russia).

## siRNA knockdown of endogenous human Eps8

The human-specific Eps8 siRNA (#s4770) used during this study was from Life Technologies. Cells were incubated with siRNA for 24 hr prior to performing experiments. Knockdown was confirmed by Western blotting of whole-cell lysates for the presence of Eps8.

## Western blotting

Whole-cell lysates were prepared by scraping cells into ice cold RIPA buffer (50 mM Hepes pH 7.4, 150 mM NaCl, 5 mM EDTA, 0.1% SDS, 0.5% deoxycholate and 1% Triton X-100) containing protease and phosphatase inhibitors (Roche, Switzerland). Before loading onto 4–12% NuPAGE Bis-Tris gradient gels (Life Technologies), lysates were cleared by centrifugation. Following SDS-PAGE, proteins in gels were transferred to nitrocellulose using an iBlot (Life Technologies). Before blocking, proteins were fixed to the nitrocellulose by air drying the membrane overnight at room temperature. Blocking of membranes was then done in blocking buffer (Hepes buffered saline containing 0.1% Triton X-100, 1% BSA, 1% fish gelatin and 5 mM EDTA). Antibodies against Eps8 (BD Biosciences, Franklin Lakes, NJ; #610143), phospho-Erk (Cell Signaling Technology #4370),Erk (Cell Signaling Technology #9102), pMLC (Rockland, Limerick, PA; #600-401-416) and MLC (Rockland #600-401-938) were used at a 1:1000 dilution and incubated overnight at 4C in blocking buffer. IRDye 680RD and 800CW secondary antibodies (LI-COR Biosciences, Lincoln, NE) were then used at 1:5000 in blocking buffer for 2 hr at room temperature after extensive washing in Hepes Buffered Saline (HBS) containing 0.1% Triton X-100. Bands were then resolved on an Odyssey scanner (LI-COR Biosciences).

## Immunoprecipitations

12 hr after transfection of plasmids encoding EGFP or Emerald-tagged mouse Eps8, A375 whole-cell lysates were prepared by scrapeing into ice cold RIPA buffer (50 mM Hepes pH 7.4, 150 mM NaCl, 5 mM EDTA, 0.1% SDS, 0.5% deoxycholate and 1% Triton X-100) containing protease and phosphatase inhibitors (Roche). Before immunoprecipitation, lysates were cleared by centrifugation. EGFP and Emerald tagged proteins were immunoprecipitated using a mouse GFP antibody (Roche #11814460001) and Protein-G magnetic beads (Life Technologies) for 2 hr at 4C. Immunoprecipitated proteins were washed 4 times with ice cold RIPA buffer before running on 4–12% NuPAGE Bis-Tris gradient gels (Life Technologies). Following SDS-PAGE, proteins in gels were electrotransferred to nitrocellulose using an iBlot (Life Technologies). For U0126 treatments, A375 cells were treated with 10 μM U0126 for 90 min and inhibition of Erk activity was confirmed by Western blotting for phospho-Erk and Erk as described under 'Western blotting.' Proteins with phosphorylated serine or threonine were detected using a rabbit anti-pS/T antibody (Abcam, United Kingdom; #ab17464) used at 1:500. Immunoprecipitated EGFP and Emerald tagged proteins was confirmed using a rabbit GFP antibody (Life Technologies #A6455) used at 1:1000.

## Immunofluorescence

Cells were cultured, stained and imaged in 6-well glass bottom dishes (In Vitro Scientific). Where noted, glass was first coated with 5 or 50 μg/ml human plasma fibronectin (Millipore) or poly-L-Lysine (Millipore) for 1 hr at 37C. Samples were fixed using 4% paraformaldehyde (Electron Microscopy Sciences, Hatfield, PA) in HBS for 20 min at room temperature. Permeabilization/blocking were performed using blocking buffer for an hour. Eps8 antibody (BD Biosciences #610143) was used at a 1:250 dilution, fluorescently conjugated phalloidin (Life Technologies) at 1:200, and fluorescently conjugated wheat germ agglutinin (Life Technologies) at 1:1000. Each was incubated in blocking buffer overnight at room temperature. Samples were gently washed several times in HBS containing 0.1% Triton X-100. An anti-mouse Alexa Fluor 488 secondary antibody (Life Technologies) was used to detect Eps8. Imaging was performed in HBS.

## Light microscopy

Immunofluorescence and time-lapse live-cell fluorescence microscopy was performed using the imaging system that is described in (*Shin et al., 2010*). Briefly, this consisted of an automated Eclipse Ti microscope equipped with the Perfect Focus System (Nikon, Japan), a servomotor driven X-Y stage with a piezo top plate (Applied Scientific Instruments, Eugene, OR) and a CSU-X1-A3 spinning disk confocal scan head (Yokogawa, Japan). Illumination for confocal imaging was provided by solid state lasers (40 mW 442 nm for CFP; 100 mW 488 nm for EGFP; 100 mW 523 nm for YFP and 500 mW 561 nm for FusionRed) that were directed to the microscope by a custom-designed optical fiber-coupled laser combiner (Spectral Applied Research, Canada; *Shin et al., 2010*). For phase contrast imaging, illumination was provided by a quartz-halogen lamp using a 546 nm bandpass filter. Images were acquired using either a CoolSNAP HQ2 or MYO cooled CCD camera (Photometrics, Tucson, AZ) using either a 100× or 60× (1.4 NA, Plan Apo PH) oil immersion objective lens and 0.9 NA condensor. Illumination, image acquisition, and microscope functions were controlled by Metamorph software (Molecular Devices, Sunnyvale, CA). For time-lapse FRET imaging of the EKAREV biosensor, a confocal image through the central plane of the cell was first acquired using 442 nm excitation and the YFP emission filter (FRET image), followed by an image acquired using 442 nm excitation and the CFP emission filter (CFP image). For photo-activation of mEos2, a Nikon A1R laser-scanning confocal microscope equipped with dual resonant scanners was used with a 60× (1.4 NA Plan Fluor) oil immersion lens. A 0.5 s pulse of the 405 nm laser at 50% power within a defined region was used to photo-convert mEos2. Images were then acquired using the 488 and 561 nm lasers in the red and green channels at 130 ms intervals to image the redistribution of fluorescence after photo-conversion. For all experiments, a stage-top incubator (Tokai-Hit, Japan) was used to maintain samples at 37C.

## Measurement of the timing of protein accumulation on bleb membranes

Cells expressing EGFP or Emerald-tagged protein were plated on uncoated 6-well glass bottom plates (In Vitro Scientific) in media containing 0.1 mg/ml rhodamine-labeled dextran (MW = 70 kDa, Sigma Aldrich). Time of arrival along the bleb perimeter was determined to be when EGFP or Emerald fluorescence was above background. Maximal protrusion of the bleb membrane as judged by displacement of rhodamine-dextran (negative stain) was set as time 0.

## Measurement of leader bleb area and percent cell migration

For determination of leader bleb area, confocal images of EGFP or Emerald through the central plane of confined cells were acquired for 4 hr at 5 min intervals. Leader bleb and cell body area were measured using images of soluble EGFP for non-targeting and hEps8 siRNA treated cells and Emerald tagged to Eps8 in rescue and over-expression experiments by outlining cells in each frame using Fiji (http://fiji.sc/Fiji). The percent of migratory cells was determined from time-lapse phase-contrast images acquired for 4 hr at 5 min intervals. The number of moving vs stationary cells was counted during the course of the video.

## Measurement of actin cortex density and thickness

Round cells on uncoated 6-well glass bottom plates were prepared as described in 'immunofluores-cence.' To measure actin cortex density, cells were stained with Alexa Fluor 568 conjugated phalloidin (Life Technologies). To measure cortex thickness, we used the methods of (*Clark et al., 2013*). Briefly, cells were double-labeled with Alexa Fluor 568-conjugated wheat germ agglutinin (Life Technologies) and Alexa Fluor 647-conjugated phalloidin. Spinning disk confocal images through the central Z-plane of the cell were acquired using a 100× (1.4 NA) objective lens. For cortex density, a 5 pixel wide line was drawn along a region of the cortex that was free of blebs and the mean fluorescence intensity was measured using Fiji (http://fiji.sc/Fiji). Additionally, background fluorescence was measured by selecting a region inside the cell. Actin cortex density was then calculated as the mean fluorescence intensity at the cortex minus background fluorescence. For determining cortex thickness, images were analyzed by performing an intensity line-scan perpendicular to and across the cell edge on a dual color image, and the distance between the peaks of the phalloidin and wheat germ agglutinin fluorescence intensity was recorded and input into the equation reported by *Clark et al. (2013)* to calculate the cortex thickness.

## Measurement of the distribution and anisotropy of proteins in leader blebs

For regional analyses of protein distribution and anisotropy, five ROIs each representing 20% of the length of the leader bleb were drawn using Fiji (http://fiji.sc/Fiji). Using FibrilTool (*Boudaoud et al., 2014*), we measured the anisotropy of F-tractin fluorescence signal within the same regions used for determining protein distribution. FusionRed-F-tractin images having similar contrast, as determined by SD/mean, were used for anisotropy analyses.

## FRET and mEos2 photo-conversion image analysis

To generate emission ratio images, we wrote a Fiji (http://fiji.sc/Fiji) macro based on previously described work (*Pertz et al., 2006*). Briefly, CFP and FRET images were first background-subtracted and corrected for bleaching by exponential fitting. CFP images were then thresholded to generate a binary mask. After multiplication by this mask, the FRET image was divided by the CFP image to yield an emission ratio reflecting Erk kinase activity throughout the cell. Ratio images were then pseudo colored using the '16 color' LUT. Similarly, images of diffusing mEos2 were pseudo colored using the '16 color' LUT.

## Atomic force microscopy

Force spectroscopy on non-adherent cells was performed using a Bioscope II AFM system (Veeco, Plainview, NY) mounted on an automated inverted epi-fluorescence microscope (Nikon Eclipse TE2000) controlled by Metamorph software (Molecular Devices). Illumination was provided by a mercury arc lamp and wavelengths selected by a Sedat filterset (Semrock, Rochester, NY). The hybrid microscope instrument was placed on an acoustic isolation table (Kinetic Systems, Boston, MA). A heating stage (Veeco) was used to maintain cells at 37C. Cells were located by EGFP fluorescence while cell radii were measured using bright-field images taken with a 40× (0.6 NA Plan Fluor) objective lens and a QuantEM EMCCD camera (Photometrics). After location of cells, a soft and tipless rectangular silicon nitride cantilever (length: 350 ± 5 μm, width: 32.5 ± 3 μm, thickness: 1 ± 0.5 μm; MikroMasch, Bulgaria) was brought to close proximity. The cantilever stiffness was first calibrated by performing a force curve on the stiffer glass-bottom dish to estimate the photodetector deflection sensitivity, and then by using the thermal noise fluctuation method (*Hutter and Bechhoefer, 1993*). The estimated spring constants for cantilevers used in force curves were 0.08–0.11 N/m. After calibration, the AFM cantilever was moved on top of a cell and lowered to gently deform it. Approximately ten successive force curves were performed in the same location on each cell per condition using 4 μm ramps with up to ~1 nN applied force at 0.5 Hz.

## Analysis of AFM data

All AFM force spectroscopy measurements were analyzed to extract the cell mechanical properties using MATLAB (Mathworks, Natick, MA). Before importing the force curves into MATLAB for analysis, each individual acquired curve was preconditioned by offsetting the y-axis to 0 and reformatted to a text file format using the NanoScope Analysis software (Bruker, Billerica, MA). We discarded noisy force curves and/or curves that presented jumps possibly due to cantilever slippage or very weakly adhered and moving rounded cells. For initial contact estimation, user-dependent determination was employed, selecting the location when the force curves increased substantially from zero. This method does not require prior knowledge or assumption about the material and geometrical properties of the cell. For fitting, Z distances between 0–400 nm were relatively consistent in yielding good fits ($R^2 > 0.9$). Curves with poor fits $R^2 < 0.9$ were discarded from the analysis.

## Determination of cortical tension and intracellular pressure

A simple theory was derived for the quantitative determination of mechanical properties of non-adherent melanoma cells confined between two flat surfaces. This will be described in detail elsewhere. Briefly, when a spherical cell is rapidly (0.5 Hz) but gently pressed down upon by the soft tipless cantilever, the cell is assumed to deform to an ellipsoid shape. The surface tension can be estimated by force balance in the vertical Z-direction. This force balance enables the relationship of the external applied cantilever force to the internal hydrostatic pressure and cortical tension. The Laplace's law describes the proportional relationship between tension and pressure. The derived expressions for the cortical tension and intracellular pressure of a non-adherent cell are:

$$T = \frac{k_c}{\pi}\left(\frac{1}{Z/d - 1}\right), \tag{S1}$$

$$P = \frac{2T}{R}, \tag{S2}$$

where $T$ is the cortical tension, $P$ is the intracellular pressure, $k_c$ is the calibrated effective cantilever spring constant, $Z$ is the Z-piezo extension distance, $d$ is the cantilever deflection and $R$ is the sample radius.

## Statistics

Statistical significance between means was determined using a two-tailed Student's t-test in GraphPad Prism (La Jolla, CA). All differences were considered significant if $p \leq 0.05$.

## Acknowledgements

We thank Bill Shin for maintenance of the Waterman lab microscopes and Schwanna Thacker for administrative assistance. We thank Ewa Paluch (UCL) for valuable discussions, Giorgio Scita (University of Milan) for providing WT and non-phosphorylatable Eps8, and Kazuhiro Aoki (Kyoto University) and Jun-ichi Miyazaki (Osaka University) for EKAREV plasmid DNA. We are grateful to the Advanced Technology Research Facility (NCI, Frederick, MD) for generating EGFP-B-Raf V600E and the NHLBI light microscopy core facility for use of the Nikon A-1R. This work was supported by funds from the intramural research program at the NIH.

## Additional information

### Competing interests

CMW: Reviewing editor for *eLife*. The other authors declare that no competing interests exist.

### Funding

| Funder | Author |
| --- | --- |
| National Heart, Lung, and Blood Institute (NHBLI) | Clare M Waterman |
| National Institute on Deafness and Other Communication Disorders (NIDCD) | Richard S Chadwick |

The funders had no role in study design, data collection and interpretation, or the decision to submit the work for publication.

### Author contributions

JSL, Conception and design, Acquisition of data, Analysis and interpretation of data, Drafting or revising the article, Contributed unpublished essential data or reagents; AXC-R, Acquisition of data, Analysis and interpretation of data; MAB, MWD, Contributed unpublished essential reagent (FusionRed-F-tractin); RSC, Conception and design, Analysis and interpretation of data; CMW, Conception and design, Analysis and interpretation of data, Drafting or revising the article

## Additional files

### Supplementary file

• Supplementary file 1. Quantitative and statistical analyses of leader bleb area, cortex tension and intracellular pressure for each condition in A375 cells. (Sheets 1–6) Quantitative and statistical analyses of leader bleb area (Sheets 1 and 2, expressed in µm²) cortex tension (Sheets 3 and 5, expressed in pN/µm) and intracellular pressure (Sheets 4 and 6, expressed in Pa) for human melanoma A375 cells treated with non-targeting siRNA (non-targeting) or depleted of Eps8 using an siRNA specific for

human Eps8 (hEps8 siRNA), rescued with or over-expressing (OE) Emerald-tagged wild type mouse Eps8 (mEps8 WT) or the following mutants: mEps8 Δbund (bundling defective, L757A/K759A), mEps8 Δcap (capping defective, V689D/L693D) and mEps8 SATA (Erk phosphorylation deficient, S624A/T628A), or EGFP-tagged human α-actinin, or treated with 50 μM blebbistatin to inhibit myosin II or 10 μM U0126 to inhibit MEK/Erk. (Sheets 1, 2) Cells were confined between uncoated glass and an agar pad, leader bleb area is expressed as percent of cell body area. In (Sheet 1), cells were depleted of Eps8 and rescued with WT and mutants of Eps8, in (Sheet 2), cells were over-expressing WT or mutant Eps8. (Sheets 3–6) Cells were plated on uncoated glass, and where noted, treated with 50 μM blebbistatin (5 min) to inhibit myosin II or 10 μM U0126 (30 min) prior to atomic force microscopy analysis. (Sheets 3, 5) Cortex tension (expressed in pN/μm) in cells (Sheet 3) depleted of and rescued with WT Eps8, or (Sheet 5) over-expressing WT Eps8 or mutants. (Sheets 4, 6) Intracellular pressure (expressed in Pa) in cells (Sheet 4) depleted of and rescued with WT Eps8, or (Sheet 6) over-expressing WT Eps8 and mutants.

---

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
