## [Decision Letter]

Thank you for submitting your work entitled “Erk regulation of actin capping and bundling by Eps8 promotes cortex tension and leader bleb-based migration” for peer review at *eLife*. Your submission has been favorably evaluated by Fiona Watt (Senior editor), and three reviewers, one of whom, Pekka Lappalainen, is a member of our Board of Reviewing Editors.

The reviewers have discussed the reviews with one another and the Reviewing editor has drafted this decision to help you prepare a revised submission.

Summary:

Animal cells can apply different mechanisms of migration depending on cell-type and physicochemical properties of the environment. Many cancer cells apply bleb-based amoeboid migration in tissue environment, but the mechanisms controlling the organization and mechanics of cortical actomyosin structures during blebbing are incompletely understood. Here Logue et al. provide evidence that actin filament bundling and capping protein, Eps8, has an important role in bleb formation. A375 cells, under confined migration, undergo a switch toward a blebbing type of locomotion characterized by the extension of a prominent large bleb that drives cells forward. Eps8 concentrates to and forms a gradient along the leader bleb, and depletion of this protein leads to defects in proper organization of actin filament bundles and bleb-based migration. RNAi-rescue and over-expression studies further demonstrated that both actin filament bundling and capping activities are important in regulation of leader bleb dynamics, and provided evidence that spatiotemporally controlled phosphorylation of Eps8 through Erk is important in coordinating these activities in cells. The data thus support a model whereby Eps8 is a major target of Erk activity in the MAT switch of migration.

The majority of the data presented in the manuscript are of good technical quality, and the work provides important new information concerning mechanisms of bleb-based migration and the role of Eps8 in regulating the cortical actin cytoskeleton to control cortex tension. This study will thus be an important contribution for the cell migration and cancer communities. However, there are few key assumptions and some experimental limitations that prevent the generalization of findings that are otherwise interesting and novel. Therefore, additional experiments and clarifications (see 'essential revisions' below) are required to further strengthen the manuscript.

Essential revisions:

1) The entire dataset was generated by using A375 melanoma cells. These cells express mutated BRAF begging the question as to whether Eps8 requirement for leader bleb formation specifically depends on this genetic alteration (that is expected to affect Erk levels). Thus, the authors should use some other cell lines to test whether Eps8 is generally required for leader bleb formation (or other related processes) or whether this is specific for A375 melanoma.

2) The authors assumed that Eps8 is phosphorylated in an Erk-dependent fashion in A375 cells, but there were no experiments to directly demonstrate this. The use of an Eps8 phospho-deficient mutant provides indirect evidence in this direction. However, it is unclear whether treatment with Erk inhibitors is effective in preventing Eps8 phosphorylation. Thus, it will be necessary to directly test if Eps8 is indeed phosphorylated in Erk-dependent fashion in A375 cells. Previously, 2D gel electrophoresis (34) was applied to provide evidence that Eps8 is indeed phosphorylated in an Erk-dependent fashion following BDNF stimulation of neurons. A similar approach might be employed here by comparing untreated A375 cells and A375 cells treated with Erk inhibitor.

3) The mechanism by which Eps8 is recruited to small blebs in 'non-confined' cells and/or to the leader bleb in 'confined' cells remains elusive. Is this because Erk is also concentrated to the leader bleb or does Eps8 just follow F-actin (or some actin-associated protein such as ezrin)? The authors should at minimum discuss this in the manuscript (unless there are no easy experiments that could be performed to address this issue).

---

## [Author Response]

*1) The entire dataset was generated by using A375 melanoma cells. These cells express mutated BRAF begging the question as to whether Eps8 requirement for leader bleb formation specifically depends on this genetic alteration (that is expected to affect Erk levels). Thus, the authors should use some other cell lines to test whether Eps8 is generally required for leader bleb formation (or other related processes) or whether this is specific for A375 melanoma*.

We tested the requirement for Eps8 in the ability of two different cancer cell lines to form leader blebs and undergo leader bleb-based migration under non-adhesive confinement. We performed experiments in human osteosarcoma U20S cells that are deleted for the cell cycle regulatory gene CDKN2A, and human lung cancer A549 cells, which are known to harbor the K-Ras G12S oncogene. siRNA depletions and rescues performed with Emerald tagged mouse Eps8 were performed in these cells lines, and we quantified leader bleb area and percent of cells that are migratory under non-adhesive confinement. The quantitative data are included in Figure 2 and images of cells are included in Figure 2—figure supplement 2. The results of these experiments indicate, first, that Walker and A549 cells undergo a morphological transition to a leader bleb morphology and exhibit leader bleb-based migration when confined against a non-adhesive coverslip under agar, and second, that normal expression level of Eps8 is required for this behavior. This demonstrates that our findings are not specific to melanoma or a single cell line, but are a more general property of cancer cells. We have added a new paragraph describing these experiments and their results in the Results section as follows: “We then sought to determine whether the requirement for Eps8 […] independent of the defect driving transformation.”

We have also added the following to the Discussion: “The Ras/Raf/MEK/Erk pathway […] lead to Eps8-dependent bleb-based migration.”

*2) The authors assumed that Eps8 is phosphorylated in an Erk-dependent fashion in A375 cells, but there were no experiments to directly demonstrate this. The use of an Eps8 phospho-deficient mutant provides indirect evidence in this direction. However, it is unclear whether treatment with Erk inhibitors is effective in preventing Eps8 phosphorylation. Thus, it will be necessary to directly test if Eps8 is indeed phosphorylated in Erk-dependent fashion in A375 cells. Previously, 2D gel electrophoresis (*[34]*) was applied to provide evidence that Eps8 is indeed phosphorylated in an Erk-dependent fashion following BDNF stimulation of neurons. A similar approach might be employed here by comparing untreated A375 cells and A375 cells treated with Erk inhibitor*.

We have performed new experiments to address this valid concern of the reviewers. As our lab is not expert in 2D gel electrophoresis, we chose to examine the effect of the Erk inhibitor U0126 on the level of serine and threonine phosphorylation (pS/T) in Eps8 constructs immunoprecipitated from A375 cells by western blot. The results of these experiments are presented in Figure 6 and clearly demonstrate that under normal conditions pS/T level is high for immunoprecipitated wild type Emerald-tagged Eps8, while treatment with U0126 abolished Erk phosphorylation on T202/Y204 in cell lysates, and reduced pS/T level in immunoprecipitated Eps8. We have added a new paragraph to the Results section that describes these experiments, the results, and conclusions (“We first exploited the highly specific inhibitor U0126 […] blocks Erk-mediated Eps8 phosphorylation.”).

*3) The mechanism by which Eps8 is recruited to small blebs in 'non-confined' cells and/or to the leader bleb in 'confined' cells remains elusive. Is this because Erk is also concentrated to the leader bleb or does Eps8 just follow F-actin (or some actin-associated protein such as ezrin)? The authors should at minimum discuss this in the manuscript (unless there are no easy experiments that could be performed to address this issue)*.

We agree with the reviewers that the mechanism of Eps8 recruitment to the cortex is an important question. First of all, we show in the manuscript that Erk is soluble and not localized to the membrane or concentrated in the leader bleb. Regarding whether Eps8 is recruited to the cortex by ezrin or actin, we had performed a few experiments to begin to address this prior to the first submission, and had begun to find that it is not as simple as one might expect based on the literature. Indeed, we are very interested in teasing this out, but feel that it is beyond the scope of the current study, and thus we choose to leave this question for the future. However, for the sake of the reviewers, we will describe the results of our preliminary experiments here.

First, we sought to determine the role of the ezrin-Eps8 interaction in recruitment of Eps8 to the bleb cortex. We utilized the ezrin Y477F mutant previously reported to disrupt the ezrin-Eps8 interaction in gut epithelial cells (51). We found that this mutant had no apparent effect on Eps8 localization to the cortex, but surprisingly we found that this mutant still immunoprecipitated with Eps8 from A375 lysates, similar to wild-type ezrin. Thus, within the context of leader bleb-based migration in cancer cells, a more detailed study will be required to understand the significance of the Ezrin-Eps8 interaction.

Second, we sought to determine the role of the Eps8-actin interaction in recruitment of Eps8 to the bleb cortex. We made the quadruple mutant L757A/K759A/V689D/L693D to inhibit both the capping and bundling activities of Eps8 (Δbund/Δcap). We tested whether these mutations affected the ability of Eps8 to bind to ezrin, and found that they had no effect on the ability of Eps8 to co-IP with ezrin. When we expressed the mutant in cells, we found that it exhibited a diffuse cytoplasmic localization and did not concentrate on the cortex. These are interesting preliminary results suggesting that Eps8 may be recruited to the cortex by binding directly to actin, however whether this is through ezrin is not possible to rule out. In the text, we state: “Our demonstration the Eps8 localizes to bleb membranes on a similar timescale as ezrin suggests that their interaction (51) may be critical to Eps8 cortical recruitment.”